# Hedgehog signaling regulates gene expression in planarian glia

Irving E Wang[1,2,3,4†], Sylvain W Lapan[1,2,3†], M Lucila Scimone[1,2,3], Thomas R Clandinin[4], Peter W Reddien[1,2,3*]

[1]Department of Biology, Massachusetts Institute of Technology, Cambridge, United States; [2]Howard Hughes Medical Institute, Massachusetts Institute of Technology, Cambridge, United States; [3]Whitehead Institute, Massachusetts Institute of Technology, Cambridge, United States; [4]Department of Neurobiology, Stanford University, Stanford, United States

**Abstract** Hedgehog signaling is critical for vertebrate central nervous system (CNS) development, but its role in CNS biology in other organisms is poorly characterized. In the planarian *Schmidtea mediterranea, hedgehog (hh)* is expressed in medial cephalic ganglia neurons, suggesting a possible role in CNS maintenance or regeneration. We performed RNA sequencing of planarian brain tissue following RNAi of *hh* and *patched (ptc)*, which encodes the Hh receptor. Two misregulated genes, *intermediate filament-1 (if-1)* and *calamari (cali)*, were expressed in a previously unidentified non-neural CNS cell type. These cells expressed orthologs of astrocyte-associated genes involved in neurotransmitter uptake and metabolism, and extended processes enveloping regions of high synapse concentration. We propose that these cells are planarian glia. Planarian glia were distributed broadly, but only expressed *if-1* and *cali* in the neuropil near *hh*[+] neurons. Planarian glia and their regulation by Hedgehog signaling present a novel tractable system for dissection of glia biology.

*For correspondence: reddien@wi.mit.edu

†These authors contributed equally to this work

**Competing interests:** The authors declare that no competing interests exist.

## Introduction

The Hedgehog (Hh) signaling pathway has been implicated in numerous developmental processes across the Metazoa, including limb and midline development in vertebrates and segmentation in *Drosophila* (*Ingham et al., 2011*). Little is known, however, about the role of Hh signaling in the Lophotrochozoa, one of the three superphyla that comprise the Bilateria. Further study and comparison with representatives of the other two Bilaterian superphyla, the Deuterostomes and the Ecdysozoa, is required to understand the evolution of this signaling pathway and its roles in metazoan biology. One member of the Lophotrochozoa, the planarian *Schmidtea mediterranea*, is a model system for the study of stem cell biology, wound response, and tissue patterning (*Reddien et al., 2005a; Sanchez Alvarado and Newmark, 1999*). Planarians are free-living platyhelminthes capable of regenerating essentially any lost tissue, a process involving the maintenance of a pluripotent stem cell population throughout adulthood (*Morgan, 1898; Reddien, 2011; Reddien and Sánchez Alvarado, 2004; Wagner et al., 2011; Witchley et al., 2013*). Inhibition of Hh signaling in planarians perturbs regeneration of the anteroposterior (AP) axis. *hh(RNAi)* animals regenerate bifurcated or no tails, whereas *ptc(RNAi)* animals regenerate anterior tails instead of heads (*Glazer et al., 2010; Rink et al., 2009; Yazawa et al., 2009*).

The planarian CNS consists of a pair of cephalic ganglia and ventral nerve cords, each comprised of a cortex of neuronal cell bodies surrounding a neurite-filled neuropil (*Morita and Best, 1965*). *hh* is expressed in cells along the medial domain of the cephalic ganglia (*Rink et al., 2009; Yazawa et al., 2009*), a location analogous to the vertebrate neural tube floor plate (*Dessaud et al.,*

*2008*). However, roles for Hh signaling in planarian nervous system regeneration have not been described, despite a wealth of information on its involvement in the CNS of other systems. The vertebrate ortholog Sonic hedgehog (SHH) is secreted from the floor plate and forms a ventral-to-dorsal morphogenetic gradient that establishes domains of transcription factor expression in the ventral neural tube (*Dessaud et al., 2008*). Each domain generates a distinct complement of progenitors that differentiate into neurons and glia (*Dessaud et al., 2008*). The *Drosophila* neurectoderm has a similar ventral-to-dorsal distribution of orthologous transcription factors, but Hh signaling is not required to establish these domains (*Cornell and Ohlen, 2000*). Hh signaling has recently been implicated in the regulation of multiple aspects of glia biology. In addition to specifying oligodendrocyte progenitors in the neural tube (*Rowitch and Kriegstein, 2010*), the pathway is also involved in inducing reactive astrogliosis in response to brain injury in adult mammals (*Sirko et al., 2013*), specifying subtypes of midline glia during *Drosophila* development (*Watson et al., 2011*), and regulating gene expression in astrocyte subtypes (*Farmer et al., 2016*). Examining the role of Hh signaling in planarian brain homeostasis and regeneration presents an opportunity to determine ancestral roles for this pathway in the differentiation and regulation of CNS cell types.

Through a tissue-specific RNA-sequencing approach, we identified two CNS-associated genes, *if-1* and *cali*, for which expression levels were strongly impacted by inhibition of *hh* and *ptc*. From analysis of the morphological and molecular features of cells expressing *if-1* and *cali* we propose that these cells are the first glial cell type to be molecularly identified in planarians. Planarian glia proximal to the Hh source express *if-1* and *cali*, whereas glia distal from the midline do not express these genes unless the Hh signaling pathway is induced by *ptc* inhibition. Therefore, we propose that the state of glia is regulated by proximity to medial Hh signaling. Our data indicate that a role for Hh signaling in regulation of CNS glia is a common feature across all three superphyla of the Bilateria.

## Results

### RNA sequencing identifies a set of CNS-enriched genes affected by inhibition of Hh

Previous results have shown that *hh* is expressed in two stripes lateral of the planarian midline, a pattern similar to the medial domain of the cephalic ganglia and ventral nerve cords (*Rink et al., 2009*; *Yazawa et al., 2009*). To determine whether *hh* is expressed in neurons, we performed double fluorescent in situ hybridization (FISH) analysis using RNA probes for *hh* and *Smed-prohormone convertase 2 (pc2)*, an established neuronal marker (*Collins et al., 2010*). Cells in the medial domain of the cephalic ganglion lobes expressed both *pc2* and *hh* (*Figure 1A*). The cholinergic neuron marker *Smed-choline acetyltransferase (chat)* (*Nishimura et al., 2010*) was also expressed in some, but not all, *hh*+ cells (*Figure 1B*).

To identify roles of Hh signaling in the planarian CNS maintenance, we examined gene expression changes using RNA sequencing of cephalic ganglia following RNAi of *hh, ptc,* or a control gene (*C. elegans unc-22*) not present in the planarian genome. We developed a dissection technique that allowed cephalic ganglia tissue to be collected from large (>2 cm) S2F1L3F2 sexual strain *S. mediterranea* animals following a brief acid-based fixation (*Figure 1C*). To test for enrichment using this dissection technique, amputated head fragments collected from CIW4 asexual strain *S. mediterranea* animals after six control dsRNA feedings were used as a reference library (*Figure 1D*). Head fragments contain cephalic ganglia as well as most major planarian tissue types (*Hyman, 1951*).

Differential expression analysis of cephalic ganglia versus head fragments following control RNAi revealed that, of the total 15,113 transcripts passing our filters (see methods), 2237 transcripts were significantly enriched and 1938 transcripts were significantly diminished in cephalic ganglia libraries over head fragment libraries (adjusted p-value <0.05, $\log_2$ fold change >1.0) (*Figure 1—figure supplement 1A*). To assess the success of our procedure in enriching CNS-associated transcripts, we examined a panel of 70 genes consisting of both experimentally validated head- and nervous system-enriched genes as well as transcripts predicted to be present in neurons based on sequence similarity to molecules with known roles in neuron biology (*Figure 1—source data 1*). Overall, members of this collection had an average $\log_2$-fold enrichment of 2.55 in cephalic ganglia tissue over head fragments, demonstrating successful enrichment of nervous system cells (*Figure 1—figure*

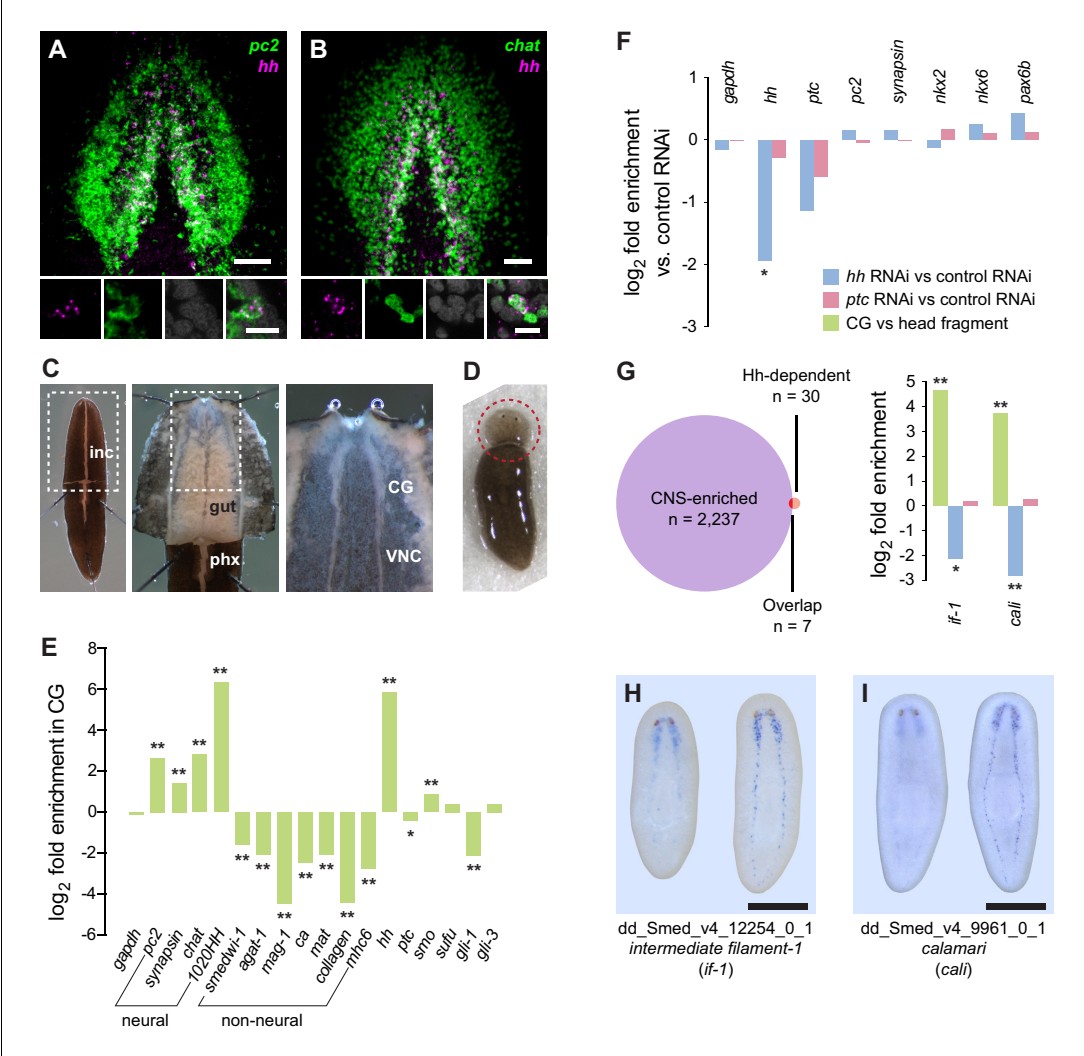

**Figure 1.** Perturbation of Hh signaling affects gene expression in the cephalic ganglia. (A–B) Double fluorescent RNA in situ hybridization (FISH) for *hh* (magenta) and neuronal markers (A) *pc2* or (B) *chat* (green) in wild-type animals. Main panels show cephalic ganglia. Lower panels show high magnification images of, from left to right, *hh* (magenta), *pc2* or *chat* (green), DAPI (gray), and merged channels from a representative double-positive neuron. (C) Excision of cephalic ganglia tissue from acid-killed animals for RNA isolation. The left panel shows incision in the dorsal epidermis. Middle panel shows detail of the boxed region in the left panel after removal of dorsal epidermis. The right panel shows the detail of the boxed region in the middle panel after removal of gut tissue overlying the cephalic ganglia and ventral nerve cords. Abbreviations: inc, incision; gut, gut branches; phx, pharynx; CG, cephalic ganglia; VNC, ventral nerve cords. See methods for dissection protocol. (D) Representative image of amputation used to collect tissue for generating the head fragment Illumina libraries. Circle indicates the portion of the animal taken for RNA isolation. (E) Bar graph depicting log$_2$ fold enrichment of selected markers in cephalic ganglia transcriptome over the head fragment transcriptome. Experimentally-verified neural markers and non-neural markers identified by brackets. Average log$_2$ fold enrichment of all 7 CNS genes listed in *Figure 1—source data 2* in cephalic ganglia transcriptome is 2.57. Average log$_2$ fold depletion of all 22 non-CNS genes listed in *Figure 1—source data 2* in cephalic ganglia transcriptome is 1.22. Statistically significant log$_2$ fold change indicated by asterisks (*p$_{adj}$≤0.05, **p$_{adj}$≤0.001). For a list of all analyzed genes, see *Figure 1—source data 1*. (F) Bar graph depicting log$_2$ fold enrichment of transcript expression level in the cephalic ganglia tissue of *hh(RNAi)* animals (blue bars) or *ptc (RNAi)* animals (red bars) over cephalic ganglia tissue from *control(RNAi)* animals. (G) Intersection of CNS-enriched genes (n = 2237) and Hh-dependent genes (n = 30) reveals 7 CNS genes misregulated following Hh pathway perturbation. Bar graph shows CNS enrichment (green bar) and relative expression following RNAi of *hh* (blue bar) or *ptc* (red bar) for *if-1* and *cali* (*p$_{adj}$≤0.05, **p$_{adj}$≤ 0.01). (H–I) WISH for (H) *if-1* and (I) *cali*. Dorsal surface shown on left, ventral surface shown on the right. Anterior up, maximum intensity projection of the ventral domain shown for A, B. Anterior up for H, I. Scale bars: 50 um for overviews, 10 um for insets for A, B; 500 um for H, I.

The following source data and figure supplements are available for figure 1:

**Source data 1.** Neuronal markers used in RNA-seq analysis and co-expression studies.

*Figure 1 continued on next page*

*Figure 1 continued*

**Source data 2.** Enrichment of neuronal markers and depletion of non-neuronal markers in cephalic ganglia tissue libraries.
**Source data 3.** Genes with significant differential expression levels following inhibition of *hh* or *ptc.*
**Source data 4.** Accession numbers of protein sequences used in phylogenetic analysis of intermediate filament proteins.
**Figure supplement 1.** Analysis of RNA-seq libraries.
**Figure supplement 2.** Hh signaling pathway perturbation does not affect regional expression of transcription factors in the central nervous system.
**Figure supplement 3.** Maximum likelihood cladogram for cytoplasmic intermediate filaments.

*supplement 1B*). Broadly expressed neuronal markers *syn, chat*, and *pc2* were only somewhat enriched (*Figure 1E*), consistent with the fact that these genes are also expressed in the peripheral nervous system located throughout the head. Conversely, genes expressed in cells restricted to the medial CNS, such as *hh* and the prohormone-encoding gene *1020HH* (*Collins et al., 2010*) were more highly enriched, at 57-fold and 81-fold, respectively (*Figure 1E*). We also examined a number of markers known to be expressed in non-neural cell types and found that whereas most of these genes were depleted in cephalic ganglia libraries, some genes were enriched (*Figure 1—source data 2*). However, the non-neuronal markers frequently used to identify specific cell types in planarians *smedwi-1, agat-1, marginal adhesive gland-1 (mag-1), carbonic anhydrase (ca), mat, collagen*, and *myosin heavy chain 6 (mhc6)* were greatly underrepresented in the CNS-specific sample (*Figure 1E*, *Figure 1—source data 2*). Genes expressed in many planarian tissues, such as *gapdh* and *ptc*, showed little difference in expression in the cephalic ganglia-versus-head dataset (*Figure 1E*, *Figure 1—source data 1*). These data indicate that although other tissues cannot be completely eliminated, the dissection protocol greatly enriches for cephalic ganglia transcripts.

To find targets of Hh signaling in the CNS, we next compared cephalic ganglia tissue from *hh (RNAi)* and *ptc(RNAi)* animals. We found insignificant differences in transcript levels for the broadly expressed housekeeping gene *gapdh* and the neural genes *syn* and *pc2* (*Figure 1F*). Expression of *hh* in *hh(RNAi)* animals was, as expected, significantly reduced ($p_{adj}$ <0.05). *ptc* expression was decreased in *ptc(RNAi)* animals as well as in *hh(RNAi)* animals (*Figure 1F*). Hh acts by negatively regulating Patched protein, which in turn is a negative regulator of transcriptional targets of Hh signaling including the *ptc* gene itself (*Varjosalo and Taipale, 2008*). Therefore, reduction of *ptc* transcript levels in *hh(RNAi)* animals was not unexpected.

The Hh signaling pathway is required for establishing expression domains of the transcription factors Nk2.2, Nk6.1, and Pax6 in the developing vertebrate neural tube (*Briscoe and Thérond, 2013*). Absence of SHH expression in the vertebrate floor plate results in loss of cell types that normally form in these domains (*Ruiz i Altaba et al., 2003*). By contrast, we were unable to find evidence that in intact planarians, which exhibit extensive tissue turnover and new cell type specification, Hh signaling modulates expression domains of orthologous transcription factors. The expression levels of *Smed-nkx2 (nkx2), Smed-nkx6 (nkx6)*, and *Smed-pax6b (pax6b)* (*Scimone et al., 2014a*) were not significantly changed in *hh(RNAi)* and *ptc(RNAi)* animals versus controls (*Figure 1F*), and we confirmed this finding by FISH (*Figure 1—figure supplement 2*).

We next conducted expression analysis for cephalic ganglia genes affected by Hh pathway perturbation. We selected a set of 30 transcripts that fit the criteria of at least 2-fold depletion or enrichment in *hh(RNAi)* or *ptc(RNAi)* samples ($p_{adj}$ <0.05), respectively, and at least 1000 RPKM to account for minor discrepancies when harvesting tissue (*Figure 1—source data 3*). Seven members of this set were CNS-enriched based on our cephalic ganglia-versus-head fragment RNA-seq data; two of these genes, *Smed-intermediate filament-1 (if-1)* and *Smed-calamari (cali)* were found to be expressed in the CNS by whole-mount in situ hybridization (WISH) (*Figure 1G–I*). *if-1* encodes a member of the cytoplasmic intermediate filament family (*Figure 1—figure supplement 3*). Intermediate filaments are cytoskeletal proteins that provide structural support and mechanical stress resistance in a variety of cell types (*Herrmann et al., 2007*). *cali* encodes a predicted protein with some

similarity to vertebrate protocadherin PCDH19 but lacks clear cadherin domains (*Frank and Kemler, 2002*), and we therefore named this gene based on the morphology of the cells expressing it (see below).

## *if-1* and *cali* expression location and levels are altered by Hh pathway perturbation

FISH analysis revealed that *if-1* and *cali* were co-expressed primarily in cells in the neuropil, the neurite-dense region surrounded by neuron cell bodies (*Best and Noel, 1969*; *Morita and Best, 1965*), of both the cephalic ganglia and the ventral nerve cords (*Figure 2A–B*, *Figure 2—figure supplement 1A*). 97.8% of *if-1$^+$/cali$^+$* cells inside the neuropil and 100% of *if-1$^+$/cali$^+$* cells outside the neuropil expressed *ptc*, indicating that these cells are likely to be responsive to Hh signaling (*Figure 2C*). Additionally, *if-1$^+$/cali$^+$* neuropil cells were adjacent to the *hh$^+$* neurons in the medial cortex, placing them in close proximity to a source of Hh ligand (*Figure 2D*). *if-1* and *cali* transcripts were detected in processes extending from the bodies of cells within the neuropil of both the cephalic ganglia and ventral nerve cords, indicating an elaborate morphology for these cells (*Figure 2E–F*). Rarely, cells expressing these two genes were also observed outside the neuropil, such as near the periphery of the head (*Figure 2G*), but the localization of these rare peripheral cells varied among animals. These isolated cells also showed high levels of expression of both *if-1* and *cali* as well as mRNA-filled processes, suggesting that they are not an artifact of the in situ hybridization protocol used.

We next assessed the impact of Hh signaling perturbation on *if-1$^+$/cali$^+$* neuropil cells. Upon *hh* RNAi, the density of *if-1$^+$/cali$^+$* cells decreased both inside and outside the neuropil (*Figure 2H–I*). Because *ptc* is a negative regulator of Hh signaling (*Varjosalo and Taipale, 2008*), we expected an increased number of *if-1$^+$/cali$^+$* cells in *ptc(RNAi)* animals. Accordingly, the density of cells expressing either or both *if-1* and *cali* in *ptc(RNAi)* animals increased slightly inside and considerably outside the neuropil (*Figure 2H–I*). Ectopic expression of these genes outside the neuropil in *ptc(RNAi)* animals was observed near the ventral surface of the animal (*Figure 2—figure supplement 1B*), with concentration of expression at the rim of the head (*Figure 2—figure supplement 1C*) near where presumptive chemosensory neurons reside (*Okamoto et al., 2005*).

During regeneration, *if-1$^+$/cali$^+$* cells accumulated in the blastema, the outgrowth that forms at wounds and replaces missing tissue. As expected, no *if-1$^+$/cali$^+$* cells were observed in the blastema in *hh(RNAi)* animals. Inhibition of *ptc* results in defective head regeneration; the cephalic ganglia in the anterior blastema appear as masses of cells without any discernable neuropil region. Nonetheless, *ptc(RNAi)* anterior blastemas had a large number of *if-1$^+$/cali$^+$* cells despite the impaired head formation (*Figure 2—figure supplement 1D*).

To ensure that ablation of *if-1* and *cali* signal resulted from loss of Hh signaling, we performed RNAi on genes encoding the planarian Gli transcription factors, which are downstream effectors of the Hh pathway (*Dominguez et al., 1996*; *Marigo et al., 1996*). *gli-1* and *gli-2*, which encode activating transcription factors, and *gli-3*, which encodes a repressing Gli-family transcription factor, have been found in the *S. mediterranea* genome (*Glazer et al., 2010*; *Rink et al., 2009*). Inhibition of *gli-1* results in a similar defective tail regeneration phenotype as does inhibition of *hh* (*Glazer et al., 2010*; *Rink et al., 2009*). RNAi of *gli-1* resulted in loss of *if-1* and *cali* signal whereas RNAi of *gli-2* and *gli-3* did not have any discernable effect on expression of *if-1* and *cali* (*Figure 2—figure supplement 2*). We conclude that Hh signaling is required for *if-1* and *cali* expression to be detected in the neuropil. Below we assess the nature of *if-1$^+$/cali$^+$* cells and whether Hh signaling regulates *if-1/cali* gene expression or the presence of these cells.

## *if-1$^+$/cali$^+$* cells are not neurons

Given the localization of *if-1$^+$/cali$^+$* cells within the CNS, we assessed whether they are neurons by examining marker gene expression. *pc2, chat,* and *syn*, three markers expressed broadly in planarian neurons, were not expressed in any *if-1$^+$/cali$^+$* cells, raising the possibility that these cells are not neurons (*Figure 3A–C*, *Figure 3—figure supplement 1A–B*). We identified and examined additional neuronal markers to further assess this possibility. Genes encoding voltage-gated ion channels, a potassium channel, a sodium channel, a calcium channel, a sodium and potassium co-transporter, Glutamic acid decarboxylase (*gd*), Tyrosine hydroxylase (*th*), Tryptophan hydroxylase (*tph*), three

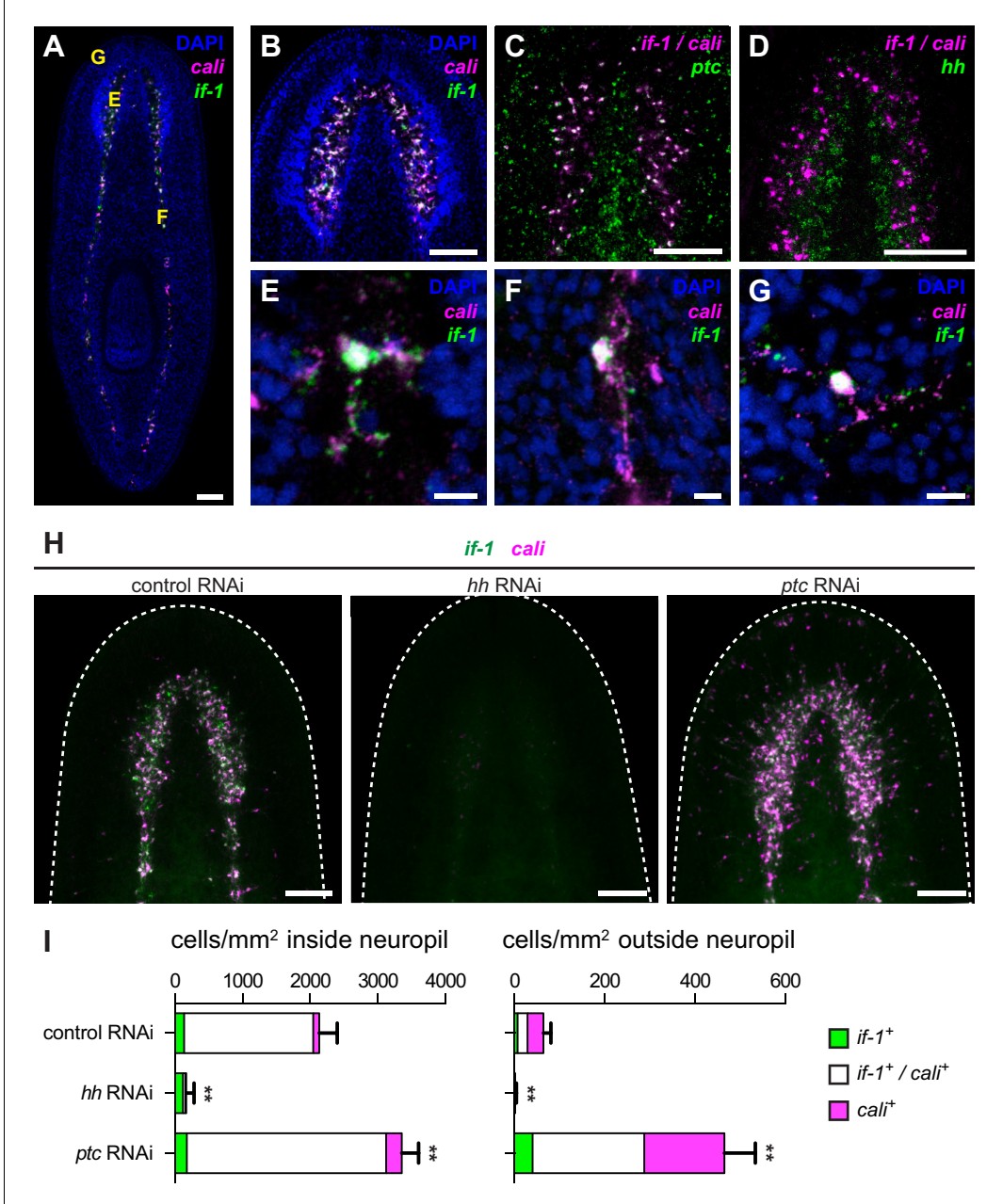

**Figure 2.** Expression of *if-1* and *cali* in neuropil cells is dependent on Hh signaling. (**A**) Double FISH for *if-1* (green) and *cali* (magenta) in wild-type animals. Cells co-expressing both markers are located in the cell body-sparse neuropil of the cephalic ganglia and ventral nerve cords. The cell body-rich cortical region is labeled by DAPI (blue). Yellow letters indicate regions detailed in E–G. (**B**) Double FISH for *if-1* and *cali* in cephalic ganglia neuropil. (**C**) Double FISH for *if-1/cali* (magenta) and *ptc* (green) indicates co-expression of the genes. Probes for *if-1* and *cali* were combined into a single channel (denoted *if-1/cali*) to improve coverage and signal intensity. 97.8 ± 2.1% of *if-1+/cali+* cells in the neuropil and 100% of *if-1+/cali+* cells outside the neuropil expressed *ptc*. (**D**) Double FISH for *if-1/cali* (magenta) and *hh* (green) indicates lack of co-expression. (**E–G**) Single *if-1+/cali+* cells in the (**E**) cephalic ganglion neuropil, (**F**) ventral nerve cord, and (**G**) head rim. (**H**) Double FISH for *if-1* (green) and *cali* (magenta) in animals following inhibition of a control gene, *hh*, or *ptc*. White dotted line delineates the edge of animal. (**I**) Quantification of the results from (**H**), with distribution of *if-1+* only cells (green), *cali+* only cells (magenta), and *if-1+/cali+* cells (white). Within the neuropil, cells expressing one or both markers are present at 2135.6 ± 265.8 cells/mm² in *control(RNAi)* conditions (n = 5 animals), 169.3 ± 118.6 cells/mm² in *hh(RNAi)* conditions (n = 4 animals), and 3354.0 ± 249.5 cells/mm² in *ptc(RNAi)* conditions (n = 5 animals). Differences were significant in both *hh* RNAi and *ptc* RNAi (**p<0.001, two-tailed t test). In the head not including the neuropil region, cells expressing one or both markers are present at 64.4 ± 16.6 cells/mm² in *control(RNAi)* conditions (n = 5 animals), 1.5 ± 2.9 cells/mm² in *hh(RNAi)* conditions (n = 4 animals), and 465.4 ± 68.7 cells/mm² in *ptc(RNAi)* conditions (n = 5 animals). Differences were significant in both *hh* RNAi and *ptc* RNAi (**p<0.001, two-tailed t test). Anterior up, ventral side shown for **A–D**, **H**. Scale bars: 100 um for **A–D**, **H**; 10 um for **E–G**.

*Figure 2 continued on next page*

*Figure 2 continued*

The following source data and figure supplements are available for figure 2:

**Source data 1.** Cell counts for *if-1* and *cali* co-expression.
**Figure supplement 1.** *if-1*[+]/*cali*[+] cells are found in multiple regions.
**Figure supplement 2.** *if-1* and *cali* expression following inhibition of gli transcription factors.

Synaptotagmin family members, Synaptogyrin 2, synaptic vesicle fusion proteins SNAP25 and Unc-13, the vesicular neurotransmitter transporters VAchT and VGluT, and neuronal transcription factors were all not expressed in *if-1*[+]/*cali*[+] cells (*Figure 3D–H*, *Figure 3—figure supplement 1C*, *Figure 1—source data 1*). *netrin-2*, a marker previously described to be expressed in cells in the neuropil (*Cebrià and Newmark, 2005*), also was not expressed in *if-1*[+]/*cali*[+] cells (*Figure 3—figure supplement 1C*). We conclude that, despite localization within the CNS and the presence of cytoplasmic extensions, *if-1*[+]/*cali*[+] cells are not neurons.

## *if-1*[+]/*cali*[+] cells express neurotransmitter reuptake and metabolism genes

In addition to neurons, the other predominant cells in the nervous systems of other organisms are glia. Glia act as neuronal support cells by providing trophic support, axon insulation, environmental maintenance, the blood-brain barrier, and synapse pruning (*Pfeiffer et al., 1993*; *Sofroniew and Vinters, 2010*). Invertebrate glia have been studied in *Drosophila* (*Hartenstein, 2011*) and *C. elegans* (*Oikonomou and Shaham, 2011*), and have been identified in annelids (*Deitmer et al., 1999*) and molluscs (*Reinecke, 1976*). Electron microscopy performed on transverse sections of the planarian *Dugesia tigrina* revealed cells distributed throughout the ventral nerve cords with lighter cytoplasmic complexity than neighboring neurons; these have been hypothesized to be planarian glial cells, but such cells had not been previously identified with molecular markers (*Golubev, 1988*; *Morita and Best, 1966*).

To determine whether *if-1*[+]/*cali*[+] cells are planarian glia, we performed FISH using RNA probes for planarian orthologs of vertebrate glia markers. Excitatory Amino Acid Transporters, which uptake the neurotransmitter glutamate from the extracellular environment (*Featherstone, 2011*), and Glutamine Synthetase, which metabolizes glutamate into glutamine (*Anderson and Swanson, 2000*), are expressed in vertebrate astrocytes (*Lehmann et al., 2009*) and *Drosophila* glia (*Soustelle et al., 2002*; *Strauss et al., 2015*). These genes act in concert to allow astrocytes to remove glutamate released during synaptic transmission and prevent excitotoxicity (*Anderson and Swanson, 2000*). *Smed-gs (gs)* encodes an ortholog of Glutamine Synthetase and was expressed in *if-1*[+]/*cali*[+] cells in the neuropil as well as in cells in the ventral parenchyma and the intestine (*Figure 3I*, *Figure 3—figure supplement 2A*). Two genes encoding orthologs of the glutamate transporter GLT-1/EAAT2 (*Featherstone, 2011*), *Smed-eaat2-1 (eaat2-1)* and *Smed-eaat2-2 (eaat2-2)* (*Figure 3—figure supplement 3*), were also expressed in the majority of *if-1*[+]/*cali*[+] cells in the neuropil and to a lesser degree outside the neuropil (*Figure 3J–K*, *Figure 3—figure supplement 2B–C*). The expression of these three genes in *if-1*[+]/*cali*[+] cells suggests a possible role in extracellular neurotransmitter clearance.

*Smed-gat (gat)* is predicted to encode an ortholog of a family of GABA, creatine, and taurine transporters that are commonly used as invertebrate and vertebrate glia markers (*Carducci et al., 2012*; *Featherstone, 2011*; *Minelli et al., 1996*; *Pow et al., 2002*) (*Figure 3—figure supplement 4*). *gat* was also expressed in *if-1*[+]/*cali*[+] cells (*Figure 3L*, *Figure 3—figure supplement 2D*). Members of the glucose transporter family are expressed in vertebrate astrocytes (*Morgello et al., 1995*; *Vannucci et al., 1997*). We found a glucose transporter ortholog, *Smed-glut (glut)* (*Figure 3—figure supplement 5*), co-expressed in *if-1*[+]/*cali*[+] cells in the neuropil as well as in cells outside the neuropil (*Figure 3M*, *Figure 3—figure supplement 2E*). Lastly, a Melastatin-Type Transient Receptor Potential Ion Channel (TRPM) ortholog was identified (*Figure 3—figure supplement 6*). In vertebrates, members of this family are expressed in oligodendrocytes (*Hoffmann et al., 2010*) and are induced

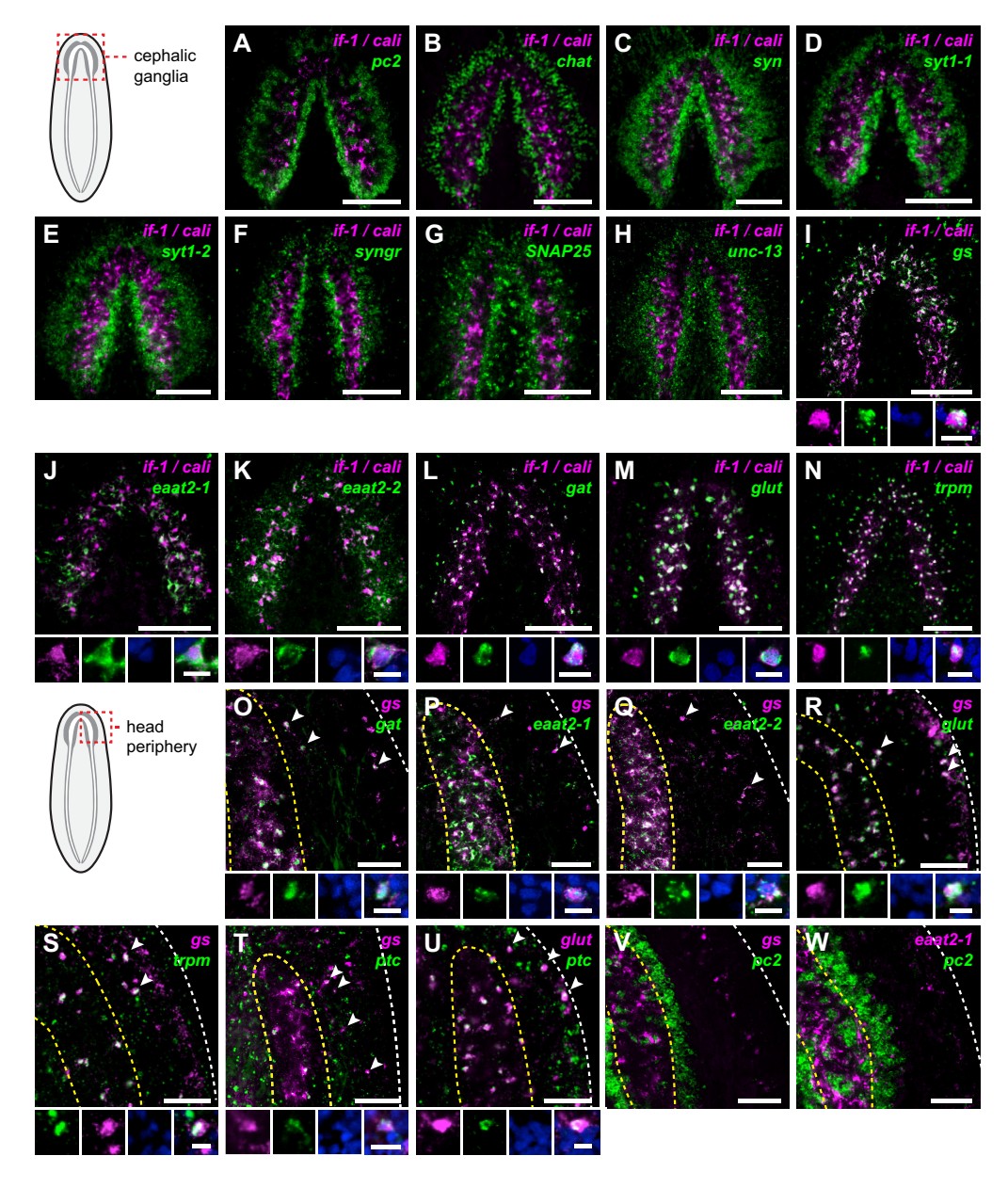

**Figure 3.** *if-1+/cali+* cells express neurotransmitter reuptake and metabolism genes. (A–N) Schematic indicates region of focus. (A–H) Double FISH of *if-1/cali* (magenta) and neural markers (A) *pc2*, (B) *chat*, (C) *syn*, (D) *syt1-1*, (E) *syt1-2*, (F) *syngr*, (G) *SNAP25*, and (H) *unc-13* (green). No co-expression was observed between neural markers and *if-1* and *cali*. (I–N) Double FISH of *if-1/cali* (magenta) with orthologs of vertebrate astrocyte markers (I) *gs*, (J) *eaat2-1*, (K) *eaat2-2*, (L) *gat*, (M) *glut*, and (N) *trpm* (green). Lower panels show high magnification images of, from left to right, *if-1/cali* (magenta), astrocyte marker ortholog (green), DAPI (blue), and merged channels from a representative double-positive cell. (O–W) Schematic indicates region of focus. The images show one hemisphere of the cephalic ganglia and the lateral parenchymal space. White dotted line delineates the edge of animal. Yellow dotted line delineates borders of the neuropil. (O–S) Double FISH of *gs* (magenta) with (O) *gat*, (P) *eaat2-1*, (Q) *eaat2-2*, (R) *glut*, and (S) *trpm* (green). 98.7 ± 1.4% of *glut+* cells in the neuropil and 99.7 ± 0.7% of *glut+* cells outside the neuropil expressed *gs*. 96.8 ± 4.6% of *trpm+* cells in the neuropil and 93.8 ± 4.5% of *trpm+* cells outside the neuropil expressed *gs*. Arrowheads denote double-positive cells outside the neuropil. Lower panels show high magnification images of, from left to right, *gs* (magenta), astrocyte marker ortholog (green), DAPI (blue), and merged channels from a representative double-positive cell. (T–U) Double FISH of *ptc* (green) with (T) *gs* and (U) *glut* (magenta). 99.3 ± 0.7% of *glut+* cells in the neuropil and 90.4 ± 4.5% of *glut+* cells outside the neuropil expressed *ptc*. Arrowheads denote double-positive cells. Lower panels show high magnification images of, from left to right, *gs* or *glut* (magenta), *ptc* (green), DAPI (blue), and merged channels from a representative double-positive cell. (V–W) Double FISH of *pc2* (green) with (V) *gs* and (W) *eaat2-1* (magenta). No double-positive cells were observed in both cases. Anterior up, ventral side shown for all.

*Figure 3 continued on next page*

*Figure 3 continued*

Maximum intensity projections shown for I–N. Scale bars: 100 um for overviews, 10 um for insets for **A–N**; 50 um for overviews, 10 um for insets for **O–W**.

The following source data and figure supplements are available for figure 3:

**Source data 1.** Cell counts for glia marker co-expression.

**Source data 2.** Accession numbers of protein sequences used in phylogenetic analysis of excitatory amino acid transporters.

**Source data 3.** Accession numbers of protein sequences used in phylogenetic analysis of GABA transporters.

**Source data 4.** Accession numbers of protein sequences used in phylogenetic analysis of glucose transporters.

**Source data 5.** Accession numbers of protein sequences used in phylogenetic analysis of transient receptor potential channels.

**Figure supplement 1.** *if-1* and *cali* expression does not overlap with neuronal marker expression.

**Figure supplement 2.** Expression patterns of markers for *if-1*$^+$/*cali*$^+$ cells.

**Figure supplement 3.** Maximum likelihood cladogram for excitatory amino acid transporters.

**Figure supplement 4.** Maximum likelihood cladogram for GABA transporters.

**Figure supplement 5.** Maximum likelihood cladogram for glucose transporters.

**Figure supplement 6.** Maximum likelihood cladogram for transient receptor potential channels.

in astrocytes during oxidative stress (*Bond and Greenfield, 2007*). The expression pattern of *trpm* was similar to that of *gs* and *gat*; *trpm* was co-expressed with *if-1* and *cali* in the neuropil, and expression was also observed in cells of the ventral parenchyma and pharynx (*Figure 3N*, *Figure 3—figure supplement 2F*).

Because we observed expression of several of these glia markers outside the neuropil, we performed double FISH analysis to determine whether these genes have overlapping expression in non-neuropil cells. Indeed, we found that *gs*$^+$ cells outside the cephalic ganglia expressed *gat* (*Figure 3O*), *eaat2-1* (*Figure 3P*), *eaat2-2* (*Figure 3Q*), *glut* (*Figure 3R*), and *trpm* (*Figure 3S*). Both *if-1* and *cali*, when ectopically expressed outside the neuropil in *ptc(RNAi)* animals, were also co-expressed with these markers (see below). Next, to determine whether this population of cells shared further similarities with *if-1*$^+$/*cali*$^+$ cells in the CNS, we examined whether these cells are responsive to Hh signaling. We found that *gs*$^+$ cells and *glut*$^+$ cells outside the neuropil also expressed *ptc*, suggesting that at least a subset of the cells are competent to respond to Hh signaling (*Figure 3T,U*). To ensure that these cells outside the neuropil were not neurons, we performed double FISH for *pc2* with *gs* or *eaat2-1* and found no evidence of co-expression (*Figure 3V,W*). Co-expression of *gs*, *eaat2-1*, *eaat2-2*, and *gat* indicates that these cells function to reuptake and metabolize neurotransmitters, a role performed in the vertebrate nervous system by astrocytes (*Anderson and Swanson, 2000*). Because these cells are embedded in the planarian nervous system and express glial markers rather than neuronal markers, we hypothesize that they are glia.

To determine the role of these glia markers in planarian biology, we performed RNAi on *gs*, *eaat2-1*, *eaat2-2*, *gat*, *glut*, and *trpm*. However, we did not observe any morphological or behavioral effects in these animals during normal tissue turnover in uninjured animals and following head and tail amputation. Inhibition of gene expression was confirmed for a subset of the glial markers by WISH analysis in RNAi animals (*Figure 3—figure supplement 2G–J*).

## IF-1 protein localizes to cellular extensions that closely associate with neurons

To examine the morphology of *if-1*[+] planarian cells, we raised a polyclonal antibody against the SMED-IF-1 (IF-1) protein. Whole-mount immunofluorescence revealed an extensive network of IF-1[+] branches concentrated in the neuropil and extending out of the CNS (*Figure 4A–B*). The IF-1[+] cellular extensions also formed hollow columns oriented along the dorsal-ventral axis (*Figure 4C*). In the periphery, IF-1[+] processes ran along tracts that were mostly devoid of cell bodies (*Figure 4D*). These peripheral branches varied between animals in extent, number, and location along the AP axis. In the VNCs, the processes ran parallel to one another (*Figure 4E*). RNAi of *if-1* resulted in complete loss of IF-1 antibody immunolabeling, confirming that labeling was specific (*Figure 4—figure supplement 1A*).

To determine whether IF-1[+] processes associated with neurons, we used antibodies against α-Tubulin and Synapsin. Antibodies against α-Tubulin label axons of both the central and peripheral nervous system in planarians (*Sanchez Alvarado and Newmark, 1999*). Axons traveling through the VNC neuropil regularly exit to form orthogonal commissures that extend from the VNC to the edge of the body. The IF-1[+] processes emerging from the cephalic ganglion neuropil followed the same tracts as the α-Tubulin[+] axon bundles (*Figure 4F*). A similar co-localization was observed in the orthogonal branches extending from the VNCs (*Figure 4G*). The IF-1[+] processes were embedded within the nerve bundles and did not appear to fully enclose the commissural axon fascicle.

An anti-Synapsin antibody labels large clusters of synapses within the neuropil and in nerve plexuses in the grid-like network of commissural axon bundles called the Orthogon (*Adell et al., 2009*; *Reisinger, 1925*; *Reuter et al., 1998*). Immunofluorescence with both the anti-IF-1 antibody and the anti-Synapsin antibody showed IF-1[+] processes weaving through the synapse-dense cephalic ganglion neuropil (*Figure 4H*, *Figure 4—figure supplement 2A*). In the ventral nerve cords, synapses accumulated into discrete, regularly spaced structures that strongly resembled synaptic glomeruli described in insect species (*Boeckh and Tolbert, 1993*). IF-1[+] processes were closely affiliated with the VNCs (*Figure 4—figure supplement 2B*) as well as along some but not all of the branches comprising the Orthogon (*Figure 4—figure supplement 2B–C*). Moreover, IF-1[+] processes appeared to encapsulate and invade Synapsin[+] clusters throughout the VNCs and the Orthogon (*Figure 4I–K*, *Figure 4—figure supplement 2D–F*). Individual IF-1[+] processes also extended from one Synapsin[+] cluster to another, indicating that single planarian glia can enwrap multiple targets (*Figure 4—figure supplement 2F*). The branched morphology of the *if-1*[+]/*cali*[+] cells and their close contact with both axons and areas of high synaptic density support our hypothesis that these cells are planarian glia that act in a similar fashion to astrocytes.

To further study the morphology of these cells, we performed protein-retention expansion microscopy (*Tillberg et al., 2016*) on animals labeled with IF-1 and Synapsin antibodies. In these animals, which have been expanded greater than 4-fold in each axis, conferring an effective lateral and axial resolution of less than 100nm to our images, IF-1[+] processes were observed forming the encompassing layer of synaptic glomeruli, with fine processes infiltrating the Synapsin[+] core (*Video 1–4*). Thinner IF-1 fibers could also be individually resolved in the VNC and the Orthogon (*Video 5–8*). These thin fibers, observed with expansion microscopy, further indicate the close association of IF-1[+] processes with regions of synaptic density.

Inhibition of *hh* resulted in complete ablation of IF-1 immunofluorescence signal and no change in expression or localization of Synapsin protein, whereas inhibition of *ptc* caused an increase in IF-1 protein presence in cellular processes observed throughout the animal (*Figure 4L*). The IF-1 protein increase observed in *ptc(RNAi)* animals manifested primarily as an increase in the number of IF-1[+] processes in contact with orthogonal axon commissures (*Figure 4—figure supplement 1B*) and at the head rim (*Figure 4M*). Normally 15.1% of orthogonal axon bundles are associated with IF-1[+] processes, whereas the percentage decreased to 2.1% following *hh* inhibition and increased to 61.4% following *ptc* inhibition (*Figure 4N*). In *ptc(RNAi)* animals, no IF-1+ processes deviated from the orthogonal axon network, which appeared normal by Synapsin labeling (*Figure 4—figure supplement 1B*). Therefore, IF-1[+] processes appear adjacent to axon bundles even when *if-1* is ectopically expressed.

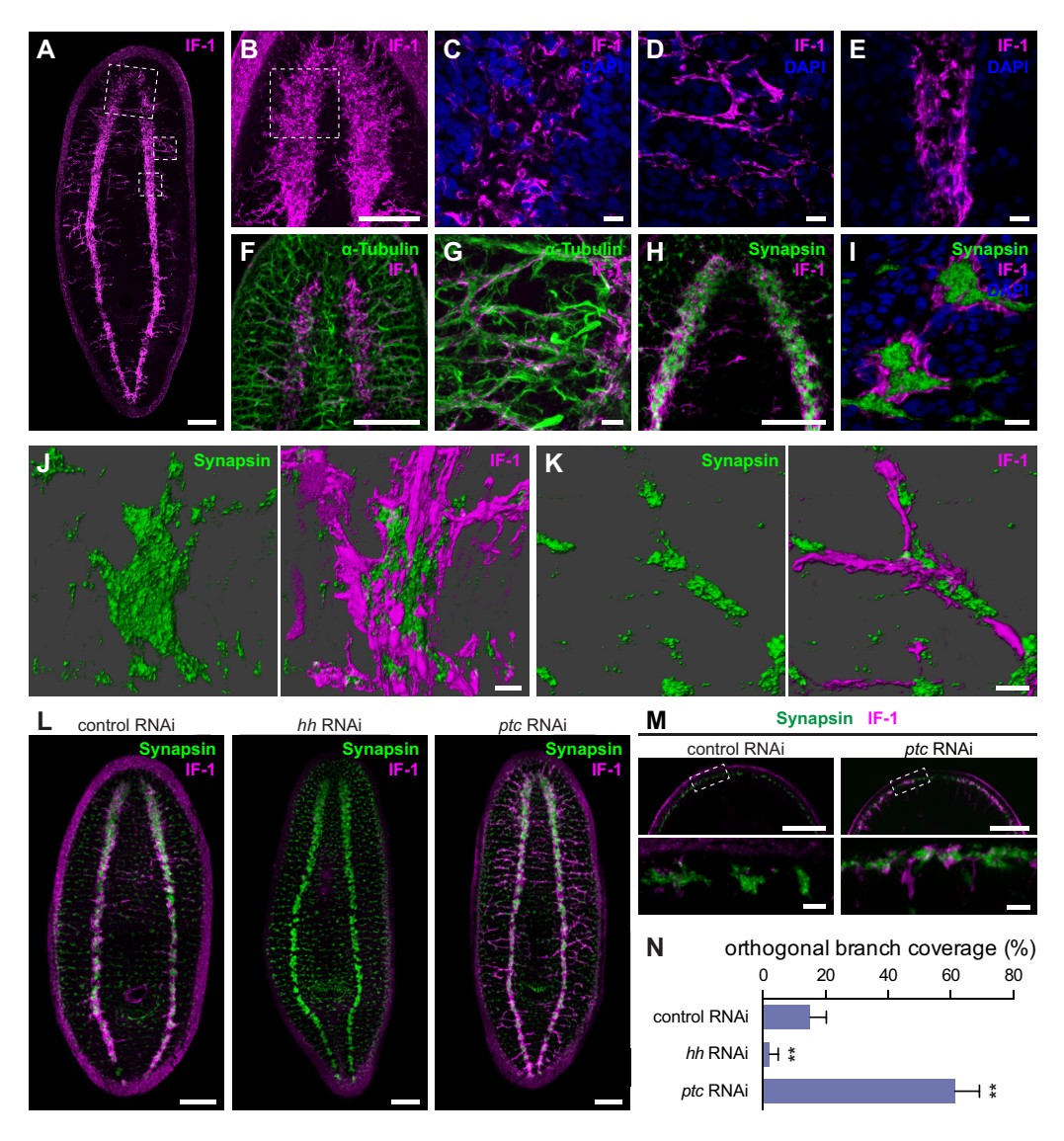

**Figure 4.** *if-1*[+]/*cali*[+] cells have processes that closely associate with neurons. (**A**) Whole-mount immunofluorescence for IF-1 protein (magenta) in wild-type untreated animals. (**B**) Maximum intensity projection of IF-1 localization (magenta) in the cephalic ganglia. Depicted region is indicated by top dotted box in panel **A**. (**C**) IF-1 localization (magenta) in the cephalic ganglion neuropil. Depicted region is indicated by dotted box in panel **B**. (**D**) IF-1 localization in the lateral ventral parenchyma. Depicted region is indicated by middle dotted box in panel **A**. (**E**) IF-1 localization in the ventral nerve cord. Depicted region is indicated by bottom dotted box in panel **A**. (**F–G**) Immunofluorescence of IF-1 (magenta) and α-Tubulin (green) in (**F**) the head and (**G**) the lateral ventral parenchyma of wild-type untreated animals. (**H–I**) Immunofluorescence of IF-1 (magenta) and Synapsin (green) in (**H**) the head and (**I**) the ventral nerve cord of wild-type untreated animals. (**J–K**) 3D renderings of confocal stacks of (**J**) a synaptic glomerulus in the ventral nerve cord or (**K**) an orthogonal branch labeled with anti-IF-1 (magenta) and anti-Synapsin (green). Image on left is Synapsin only and image on right is Synapsin and IF-1. (**L**) Immunofluorescence of IF-1 (magenta) and Synapsin (green) following inhibition of *hh*, *ptc*, or a control gene. (**M**) Detail of immunofluorescence of IF-1 (magenta) and Synapsin (green) in the head rim of animals following inhibition of a control gene or *ptc*. Dotted box in top row refers to the corresponding image in the bottom row. (**N**) Quantification of *hh(RNAi)* and *ptc(RNAi)* phenotypes based on percentage of orthogonal axon bundles in contact with IF-1[+] processes. In *control(RNAi)* animals, 15.1 ± 5.1% of orthogonal axon bundles contained IF-1[+] processes (n = 5 animals). In *hh(RNAi)* animals, 2.1 ± 2.8% of orthogonal axon bundles contained IF-1+ processes (n = 5 animals). In *ptc(RNAi)* animals, 61.4 ± 7.8% of orthogonal axon bundles contained IF-1[+] processes (n = 4 animals). The differences between both *hh* RNAi and *ptc* RNAi vs control were statistically significant (**p<0.001, two-tailed t test). Anterior up, ventral side shown for all. Scale bars: 100 um for **A**, **B**, **F**, **H**, **L**, top row of **M**; 10 um for **C**, **D**, **E**, **G**, **I**, **J**, **K**, bottom row of **M**.

The following source data and figure supplements are available for figure 4:

**Source data 1.** Orthogonal branch coverage counts following Hh pathway perturbation.

*Figure 4 continued on next page*

*Figure 4 continued*

**Figure supplement 1.** IF-1 protein accumulates in *ptc(RNAi)* animals.

**Figure supplement 2.** IF-1 protein-containing processes associated with Synapsin[+] clusters.

## Inhibition of hh does not result in loss of planarian glia

To determine whether *if-1*[+]/*cali*[+] cells in the neuropil represent a glial subtype that requires constitutive Hh signaling for survival, we examined the persistence of IF-1 protein in *hh(RNAi)* animals. If Hh signaling were required for the survival of *if-1*[+]/*cali*[+] cells, then we would expect to see the loss of both *if-1* mRNA and IF-1 protein when the cells die. Conversely, if inhibition of Hh signaling affected transcription of *if-1* but did not impact survival of these cells, then we would expect IF-1 protein to perdure for some time after *if-1* mRNA is lost. We performed a shortened RNAi treatment (3 feedings at 4-day intervals) because IF-1 protein is completely eliminated by full treatment (*Figure 4L*). In *hh(RNAi)* animals we observed IF-1 protein by immunofluorescence despite the loss of detectable *if-1* FISH signal throughout the neuropil, suggesting that *hh* RNAi impacts *if-1* and *cali* transcription in existing glia over the 12-day period during which we performed RNAi (*Figure 5A*).

To further investigate the role of Hh signaling in glia biology we examined the expression of *gs, gat, eaat2-1*, and *eaat2-2* in uninjured *hh(RNAi)* and *ptc(RNAi)* animals. Unlike the case for *if-1* and *cali*, expression of the other glia markers was still observed throughout the neuropil and was indistinguishable from control animals. Similarly, inhibition of *ptc* had no effect on the expression or localization of *gs, gat, eaat2-1*, or *eaat2-2* (*Figure 5B*). We also did not observe a significant change in the total number of *gat*[+] cells within the neuropil in *hh(RNAi)* and *ptc(RNAi)* animals compared to control animals, although the proportion of the *gat*[+] population that co-expressed *if-1* and *cali* was reduced in *hh(RNAi)* animals (*Figure 5C*). Conversely, whereas a small proportion of *gat*[+] glia outside the

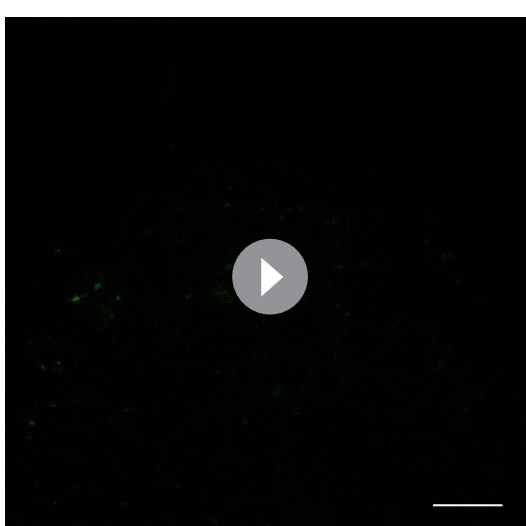

**Video 1.** Confocal stack of single synaptic glomerulus. Immunofluorescence for IF-1 (magenta) and Synapsin (green) followed by protein-retention expansion microscopy. Anterior up, ventral side shown. Scale bar: 50 um.

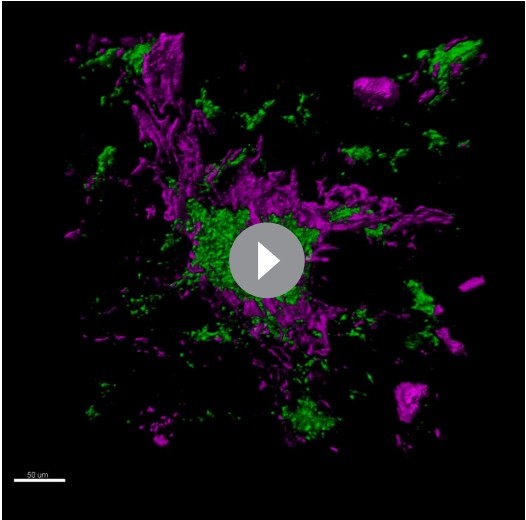

**Video 2.** 3D rendering from confocal stack of single synaptic glomerulus. Immunofluorescence for IF-1 (magenta) and Synapsin (green) followed by protein-retention expansion microscopy. 3D rendering based on confocal stack from *Video 1*. Anterior up, ventral side shown. Scale bar: 50 um.

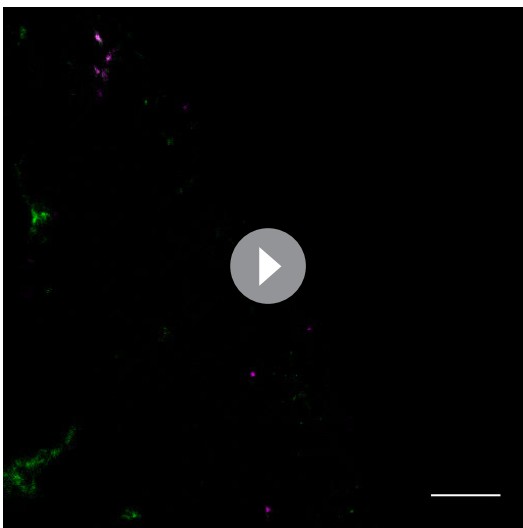

**Video 3.** Confocal stack of multiple synaptic glomeruli. Immunofluorescence for IF-1 (magenta) and Synapsin (green) followed by protein-retention expansion microscopy. Anterior up, ventral side shown. Scale bar: 50 um.

neuropil expressed *if-1* and *cali* in *control(RNAi)* animals, a large number of *gat*+/*if-1*+/*cali*+ cells were detected outside the neuropil in *ptc(RNAi)* animals despite no significant overall increase in the number of *gat*+ cells (*Figure 5C*). Given that planarian glia outside the neuropil also expressed *ptc*, these data suggest that *if-1* and *cali* were induced in these cells when Hh signaling was activated by *ptc* inhibition. We also examined transcript abundance of glia markers in our RNA-Seq data and found no statistically significant differential expression for *eaat2-1*, *eaat2-2*, *gs*, *gat*, *glut*, and *trpm* following *hh* or *ptc* RNAi (*Figure 5—figure supplement 1*). These results indicate that *if-1* and *cali* expression is lost in a population of *gs*+ and *gat*+ cells when *hh* is inhibited.

We next assessed whether planarian glia can be formed during regeneration in *ptc(RNAi)* and *hh(RNAi)* animals. Anterior blastemas of *control (RNAi)* animals after six days of regeneration contained cells expressing planarian glia markers, both inside the forming neuropil and outside. Similar results were observed in *ptc (RNAi)* animals, despite defective head formation (*Figure 5—figure supplement 2*). In *hh (RNAi)* animals, expression of *if-1* and *cali* was eliminated, but cells expressing *gat, eaat2-1*, and *eaat2-2* were observed throughout the blastema (*Figure 5—figure supplement 2*). The presence of these markers in newly formed cells of the blastema suggests that the animal is capable of regenerating glia in the absence of Hh signaling.

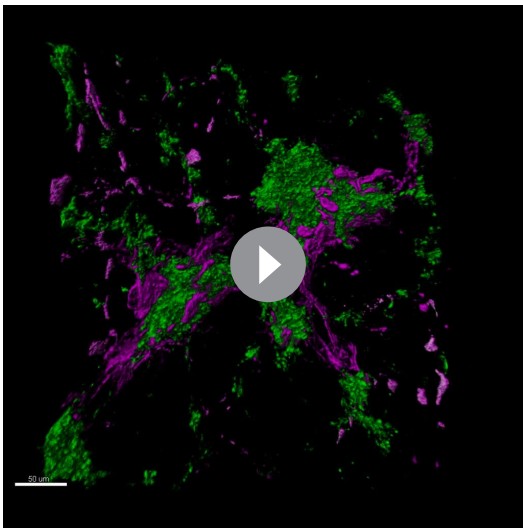

**Video 4.** 3D rendering from confocal stack of multiple synaptic glomeruli. Immunofluorescence for IF-1 (magenta) and Synapsin (green) followed by protein-retention expansion microscopy. 3D rendering based on confocal stack from *Video 3*. Anterior up, ventral side shown. Scale bar: 50 um.

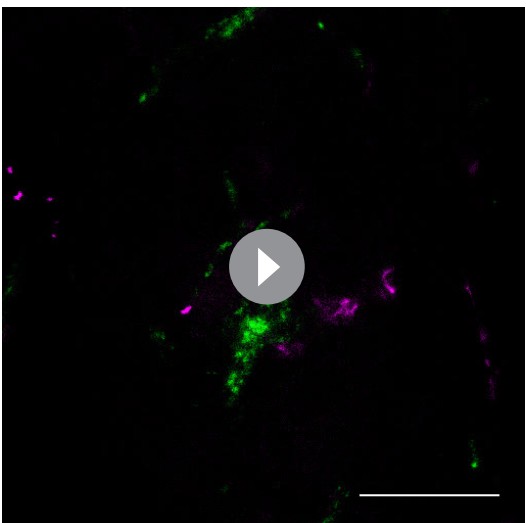

**Video 5.** Confocal stack of lateral orthogonal branch. Immunofluorescence for IF-1 (magenta) and Synapsin (green) followed by protein-retention expansion microscopy. Anterior up, ventral side shown. Scale bar: 50 um.

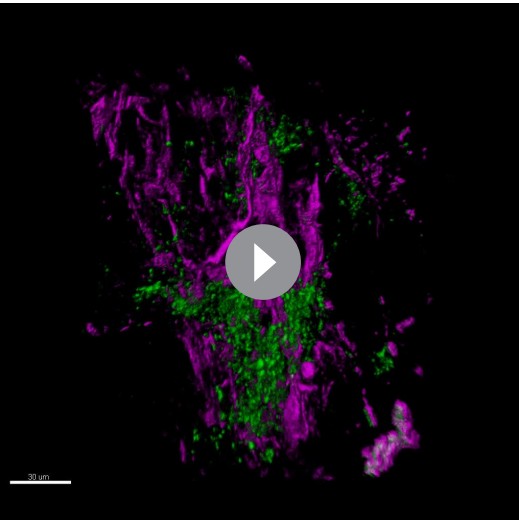

**Video 6.** 3D rendering from confocal stack of lateral orthogonal branch. Immunofluorescence for IF-1 (magenta) and Synapsin (green) followed by protein-retention expansion microscopy. 3D rendering based on confocal stack from *Video 5*. Anterior up, ventral side shown. Scale bar: 30 um.

**Video 7.** Confocal stack of medial orthogonal branch. Immunofluorescence for IF-1 (magenta) and Synapsin (green) followed by protein-retention expansion microscopy. Anterior up, ventral side shown. Scale bar: 50 um.

## Hh signaling promotes expression of *if-1* and *cali* in existing glia

The effect of Hh signaling on *if-1* and *cali* expression could occur dynamically in mature glia cells, or could exist only during the formation of neuropil glia that subsequently express *if-1* and *cali*. To distinguish between these two possibilities, we examined whether ectopic *if-1* and *cali* expression in *ptc(RNAi)* animals required new cell production. After irradiation, animals can survive for a short time but are unable to produce new cells (*Reddien et al., 2005a*; *Wolff and Dubois, 1948*). We exposed animals to 6000 rads of ionizing radiation and subsequently began RNAi. If ectopic *if-1* and *cali* expression resulting from *ptc* inhibition required new cell production, then we would expect to see no or reduced ectopic *if-1*[+]/*cali*[+] cells outside the neuropil in irradiated *ptc(RNAi)* animals. By contrast, we observed an increase in the number of *if-1*[+]/*cali*[+] cells in *ptc (RNAi)* animals despite irradiation (*Figure 6A–C*). The number of cells that expressed *if-1* and *cali* in irradiated *ptc(RNAi)* animals was similar to the results described above in unirradiated animals (*Figure 2H–I*). The ectopic *if-1*[+]/*cali*[+] cells outside of the neuropil in irradiated *ptc(RNAi)* animals had branches, indicating that ectopic expression occurred in existing cells with complex morphology (*Figure 6B*). Furthermore, the total number of *glut*[+] cells outside of the neuropil was similar in *control(RNAi)* and *ptc(RNAi)* irradiated animals, but the proportion of *glut*[+] cells that expressed *if-1* and *cali* was higher following RNAi of *ptc* (*Figure 6D*). This indicates

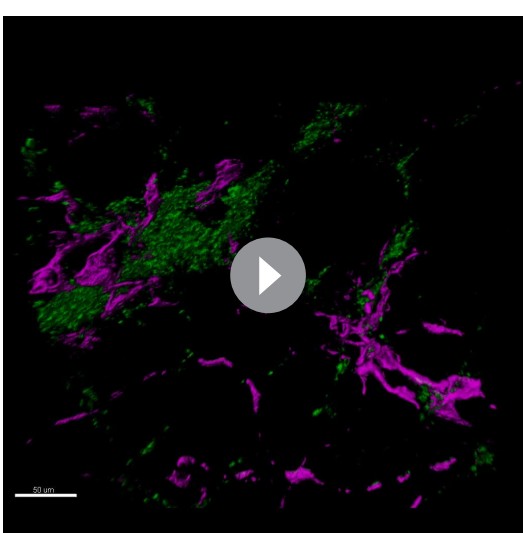

**Video 8.** 3D rendering from confocal stack of medial orthogonal branch. Immunofluorescence for IF-1 (magenta) and Synapsin (green) followed by protein-retention expansion microscopy. 3D rendering based on confocal stack from *Video 7*. Anterior up, ventral side shown. Scale bar: 50 um.

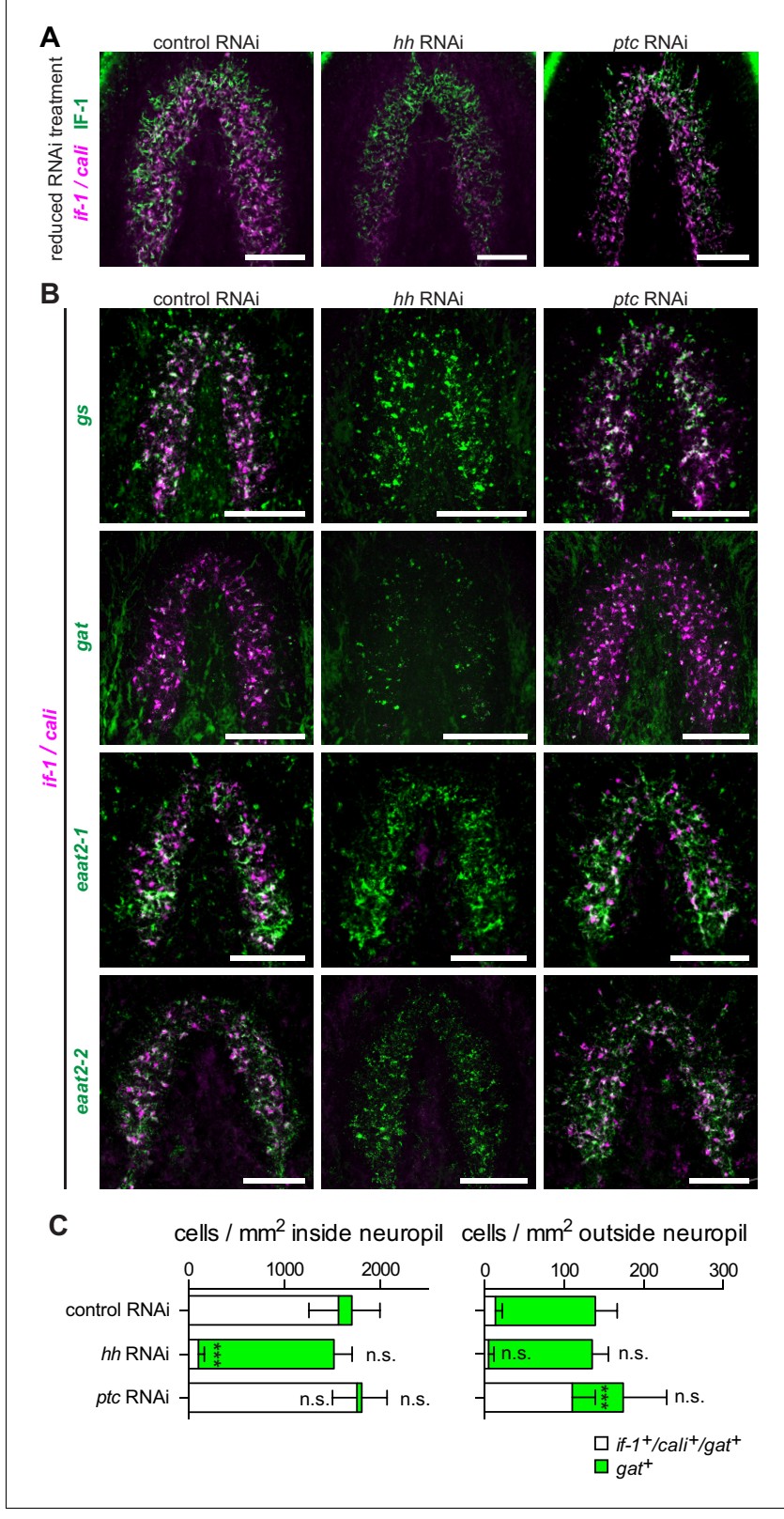

**Figure 5.** *hh* inhibition does not ablate planarian glia. (**A**) FISH of *if-1/cali* (magenta) and immunofluorescence of IF-1 (green) in animals following reduced RNAi treatment (fed d0, d4, d8, fixed d12) of a control gene, *hh*, or *ptc*. (**B**) Double FISH for *if-1/cali* (magenta) with *gs* (first row), *gat* (second row), *eaat2-1* (third row), or *eaat2-2* (fourth row) (green) following inhibition of a control gene (first column), *hh* (second column), or *ptc* (third column). (**C**) *Figure 5 continued on next page*

*Figure 5 continued*

Stacked bar graph of the number of cells per square millimeter of cephalic ganglia inside the neuropil (left) and number of cells per square millimeter of the head outside the neuropil (right) expressing *gat* following inhibition of a control gene, *hh*, or *ptc*. Bar sections denote the ratio of *if-1*[+]/*cali*[+] subpopulation (white) to *if-1*[-]/ *cali*[-] subpopulation (green). Statistical significance indicated by labels (n.s., not significant, ***p ≤ 0.0001, two-tailed t test). Anterior up, ventral side shown for **A–B**. Scale bars: 100 um for **A–B**.

The following source data and figure supplements are available for figure 5:

**Source data 1.** Cell counts for co-expression of *if-1/cali* and *gat* following Hh pathway perturbation.
**Figure supplement 1.** Glial marker expression levels in RNA-seq datasets.
**Figure supplement 2.** Expression of glial markers in anterior blastemas.

that *ptc* RNAi induced expression of *if-1* and *cali* in existing *glut*[+] cells. *glut*[+] cells were not overtly irradiation sensitive (*Figure 6—figure supplement 1*), and therefore are likely mature cells rather than progenitors. These observations indicate that cells with ectopic *if-1* and *cali* expression in *ptc (RNAi)* animals are likely mature planarian glia, and support a model that Hh signaling normally induces expression of *if-1* and *cali* in planarian glia dependent on their proximity to *hh*[+] neurons.

## Discussion

### Evidence for planarian glia

Previous electron microscopy studies had identified candidate planarian glia based on their localization and appearance but did not provide any molecular evidence for their identity (*Golubev, 1988*; *Morita and Best, 1966*). We have described here the first molecular and morphological evidence for neuronal support cells in planarians. First, the greatest accumulation of planarian glia expressing orthologs of glia markers is in the neuropil, a region filled with axons (based on α-Tubulin immunofluorescence) and synapses (based on Synapsin immunofluorescence). Second, the cells have branched processes that are closely associated with neurons. These processes extend through the synapse-rich regions of the neuropil, travel along orthogonal commissures of the peripheral nervous system, and encapsulate synaptic glomeruli. Third, these cells express three neurotransmitter transporters. Orthologs of the proteins encoded by planarian *eaat2-1* and *eaat2-2* have known roles in the transport of glutamate from the extracellular environment into the cytoplasm where it is metabolized by orthologs of the enzyme encoded by *glutamine synthetase* (*Anderson and Swanson, 2000*), another gene expressed in these planarian cells. Glutamate released from the pre-synaptic neuron, if not removed from the synaptic cleft, can continue to activate glutamate receptors on the post-synaptic neuron, resulting in high intracellular levels of calcium and activation of pathways that lead to cellular damage (*Manev et al., 1989*). Additionally, because of the expression of *gat*, which encodes an ortholog of a GABA transporter, we predict that this cell type is also involved in GABA reuptake. Based upon these data, we propose that these planarian cells uptake the excitotoxic neurotransmitter glutamate from areas near synapses to prevent damage to the nervous system, similar to the function of astrocytes in other animals (*Schousboe, 2003*). Taking these data together, we propose that these planarian cells are glia. Continued study of the function, morphology, and molecular characteristics of these cells will allow further comparison of similarities and differences between these cells and glia in other organisms. These glia markers are also co-expressed in cells outside the neuropil region, indicating the presence of glia in the nervous system beyond the neuropil of the cephalic ganglia and ventral nerve cords. The specific function of the *if-1*[+]/*cali*[+] glia within the neuropil remains to be determined. One hypothesis is that they are specialized to modulate environments of extremely high synaptic density, particularly around synaptic glomeruli that are characteristic of this region of the CNS.

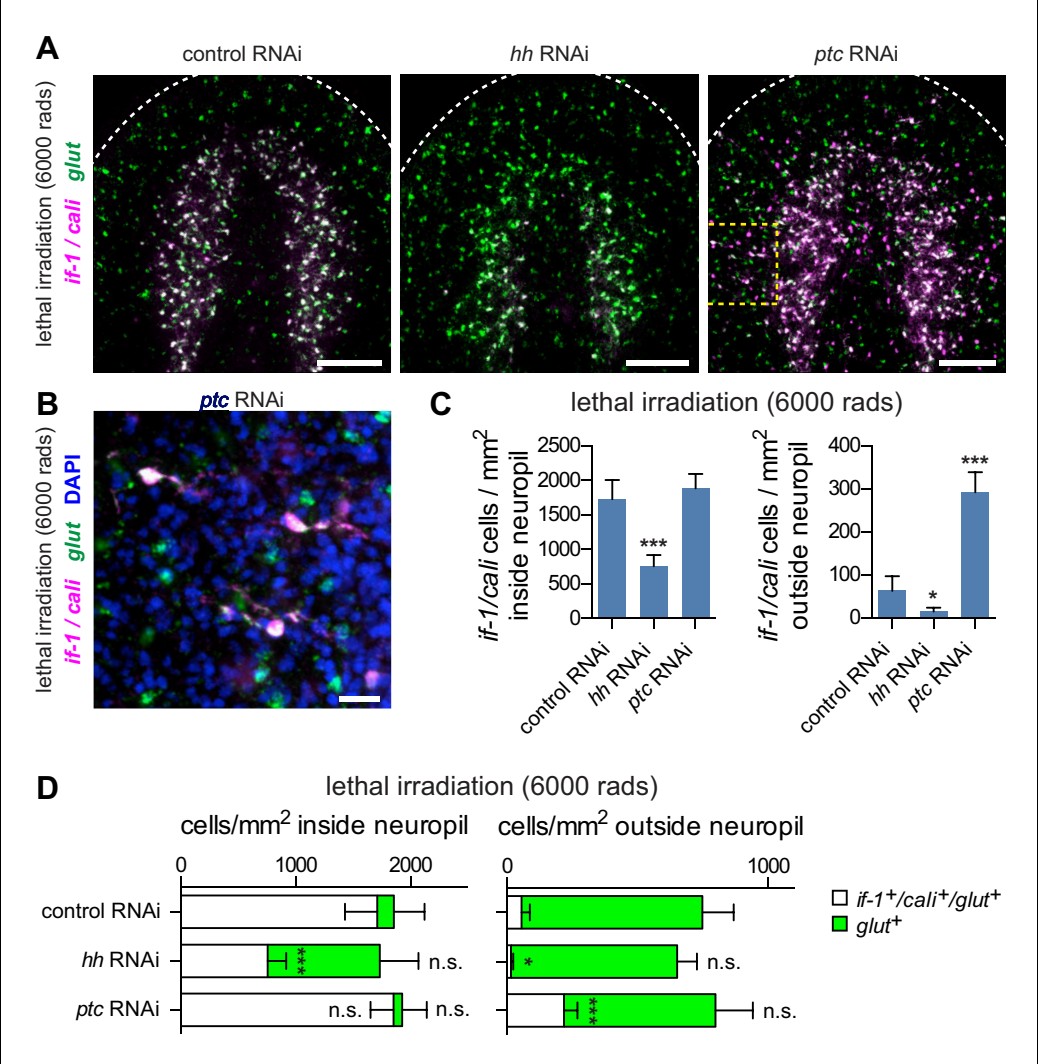

**Figure 6.** Hh signaling is required for *if-1* and *cali* expression in planarian glia. (**A**) FISH of *if-1/cali* (magenta) and *glut* (green) in animals following lethal irradiation and subsequent RNAi treatment (irradiated d0, fed d0, d4, d8, fixed d11). Yellow boxed area indicates region detailed in **B**. (**B**) Detail of *if-1*$^+$/*cali*$^+$/*glut*$^+$ cells in the head region lateral to cephalic ganglia in *ptc(RNAi)* animal. (**C**) Quantification of *if-1*$^+$/*cali*$^+$ cells in irradiated *control(RNAi)*, *hh* (*RNAi*), and *ptc(RNAi)* animals from (**A**). *Control(RNAi)* animals had 1730.29 ± 274.57 cells/mm$^2$ inside and 63.51 ± 33.93 cells/mm$^2$ outside the neuropil (n = 8 animals). *hh(RNAi)* animals had 758.51 ± 160.33 cells/mm$^2$ inside and 15.02 ± 9.10 cells/mm$^2$ outside the neuropil (n = 8 animals). *ptc(RNAi)* animals had 1888.48 ± 206.34 cells/mm$^2$ inside and 293.20 ± 46.69 cells/mm$^2$ outside the neuropil (n = 9 animals). Differences between *control(RNAi)* and *hh(RNAi)* animals (*p<0.05, two-tailed t test), and between *control(RNAi)* and *ptc(RNAi)* animals (***p<0.0001, two-tailed t test) are significant. (**D**) Stacked bar graph of the number of cells per square millimeter of cephalic ganglia inside the neuropil (left) and number of cells per square millimeter of head outside the neuropil (right) expressing *glut* following inhibition of a control gene, *hh*, or *ptc*. Bar sections denote ratio of *if-1*$^+$/*cali*$^+$subpopulation (white) to *if-1*$^-$/*cali*$^-$subpopulation (green). Statistical significance indicated by labels (n.s., not significant, *p≤0.05, ***p≤0.0001, two-tailed t test). Anterior up, ventral side shown for all. Scale bars: 100 um for **A**, 20 um for **B**.

The following source data and figure supplement are available for figure 6:

**Source data 1.** Cell counts for co-expression of *if-1/cali* and *glut* following lethal irradiation and Hh pathway perturbation

**Figure supplement 1.** Expression of glut is not affected by lethal irradiation.

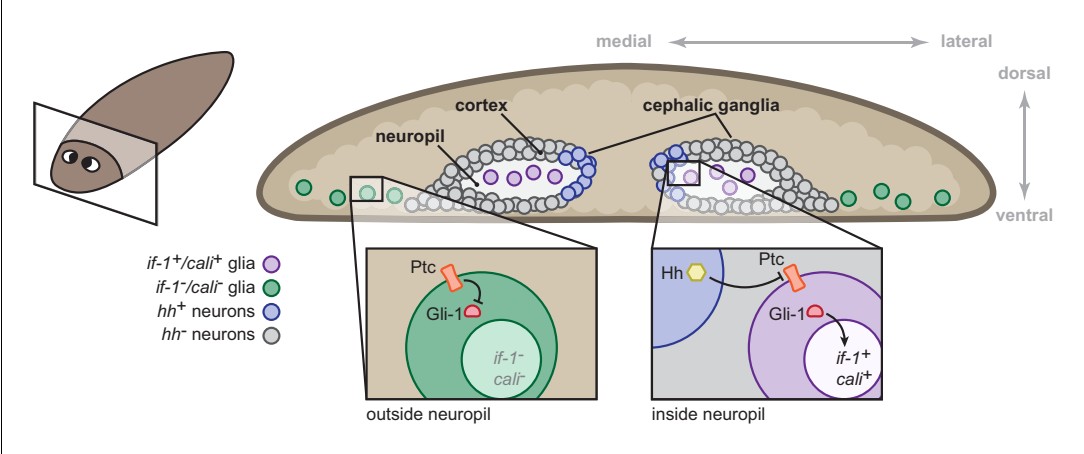

**Figure 7.** Model for the role of the Hh signaling pathway in regulation of planarian glia. Planarian glia are distributed throughout the nervous system and concentrated in the neuropil. *if-1* and *cali* expression is normally repressed in these cells by high Ptc activity. A subset of these cells, however, is adjacent to Hh-secreting neurons in the medial cortex and express *if-1* and *cali* because of inhibition of Ptc by Hh. Upon global inhibition of *hh* by RNAi, Ptc remains high in all cells and *if-1* and *cali* are repressed in glia. Upon global inhibition of *ptc* by RNAi, expression of *if-1* and *cali* is derepressed in glia.

## Hh signaling regulates gene expression in planarian glia

During homeostasis, constitutive expression of *hh* is required for expression of *if-1* and *cali* in planarian glia in the neuropil. Upon inhibition of *hh*, these cells cease transcription of *if-1* and *cali*. Inhibition of *ptc* results in ectopic *if-1* and *cali* transcription in cells distributed broadly in the animal, likely as a consequence of derepression of the Gli-1 transcription factor. This indicates that cells competent to respond to Hh ligand normally exist outside of the medial CNS. Additionally, the accumulation of *if-1* and *cali* transcripts in cells outside the neuropil in lethally irradiated (blocking all new cell production) *ptc(RNAi)* animals demonstrates that Hh signaling induces expression of the two genes in additional existing cells. Our data suggest that ectopic *if-1* and *cali* expression in *ptc(RNAi)* animals occurs in the normally *if-1⁻/cali⁻* planarian glia outside the neuropil (**Figure 7**). We currently do not have evidence supporting or rejecting the presence of multiple planarian glia cell types. It will be of interest to further investigate these possibilities by examining the function of planarian glia responsive to Hh signaling as well as planarian glia outside of the neuropil that do not express *if-1* or *cali* in *ptc(RNAi)* animals.

The ability of Hh signaling to modulate function in glia has been described in vertebrates. In reactive astrogliosis, the mammalian CNS response to injury, SHH is one of the inductive signals that induces expression of the intermediate filament GFAP (*Sirko et al., 2013*). Increased levels of GFAP protein result in an increase in cell size, which is necessary for the formation of an astrocytic scar at the wound site (*Wilhelmsson et al., 2004*). Although the function of this regulation has differences, the regulation of intermediate filament proteins by Hh signaling in glia is another striking commonality between planarians and vertebrates. Additionally, SHH secreted from neurons has been reported to regulate distinct subpopulations of mammalian astrocytes (*Farmer et al., 2016*; *Garcia et al., 2010*). Although the genes regulated by Hh signaling in these glial populations is different, the ability for neurons to instruct astrocyte expression profiles in mammals is a strikingly similar feature to what we observe in planarians (*Farmer et al., 2016*).

## Ancestral roles of Hh signaling in CNS development

Hh signaling plays a critical role in the vertebrate CNS pattern formation but a seemingly less direct role in *Drosophila*. SHH expression in the vertebrate floor plate establishes distinct domains of transcription factor expression in the ventral neural tube. These domains first give rise to neurons and then, at later stages of development, glia (*Dessaud et al., 2008*; *Yu et al., 2013*). The dorsal-ventral distribution of transcription factors in the developing CNS of *Drosophila* and the Lophotrochozoan *Platynereis dumerilii* bear a resemblance to the distribution of orthologous transcription factors in

the vertebrate neural tube (*Cornell and Ohlen, 2000*; *Denes et al., 2007*). Hh, however, appears to play a role in the anterior-posterior patterning of *Drosophila* neuroblasts rather than dorsal-ventral patterning (*Bhat, 1999*). Similarly, in *Platynereis*, a role for Hh in segment patterning has been identified, but no effect of pathway perturbation on the dorsal-ventral (medial-lateral) arrangement of CNS progenitor domains was described (*Denes et al., 2007*).

Here, we also find that regionalized expression in the cephalic ganglia of several orthologs of Hh-responsive vertebrate neural tube transcription factors appear unaffected by Hh signaling in uninjured planarians. This is consistent with the possibility that Hh signaling was co-opted into a dorsal-ventral patterning role in the nervous system in the deuterostome lineage. The lack of head formation in *ptc(RNAi)* animals is a challenge for investigating head patterning in regenerating planarians, and further research into the neuronal progenitor pool in head blastemas following Hh pathway perturbation will be of interest for continuing to assess whether any role of Hh signaling exists in planarian nervous system patterning.

The floor plate, which is induced by SHH secreted from the notochord, serves as a mediator of axonal midline crossing through the secretion of axon guidance cues (*Colamarino and Tessier-Lavigne, 1995*). SHH continues its involvement in neural patterning by acting as a chemoattractant and by mediating cellular responses to other guidance cues (*Parra and Zou, 2010*). The *Drosophila* midline glia are considered to be an analogous structure to the vertebrate floor plate because of similar gene expression and roles in controlling midline crossing (*Evans and Bashaw, 2010*). Hh in *Drosophila* is required for the decision to form posterior midline glia, the function for which is still not fully understood, instead of anterior midline glia, which develop into ensheathing glia in the *Drosophila* neuropil (*Watson et al., 2011*). A shared function of Hh signaling among Deuterostomes, Ecdysozoans, and Lophotrochozoans therefore appears to be in the control of glia near the midline.

## Implications of molecular evidence for planarian glial cells

Planarians are an ideal model for the study of regeneration because of their nearly unrivaled regenerative ability, their ease of culture, and the molecular tools developed for rapid study of gene function. The role of glia in regeneration has been investigated in vertebrates, where glia proliferate in response to brain injury, and in insects, where surface glia can reform the blood-brain barrier (*Sofroniew, 2009*; *Treherne et al., 1984*). Interestingly, astrocytic scars appear to counteract neural regeneration by blocking the extension of axons into the damaged region (*Silver and Miller, 2004*). Whether planarian glia actively participate in repatterning the nervous system after injury is an interesting topic to explore, possibly leading to studies on both mechanisms of glia-neuron interaction and glial roles in neural network connectivity. If, on the other hand, planarian glia passively extend their processes into existing neural architecture, then the mechanisms that guide glial cell development and migration could be studied instead. Several lines of evidence support the second hypothesis: IF-1$^+$ processes are not seen deviating from axonal tracts and perturbations affecting gene regulation in planarian glia do not result in observable disruption to the neural network.

The work we present here opens the field to a number of opportunities for continued research. Glia are now gaining recognition as an active player in nervous system development, function, and regeneration (*Freeman and Rowitch, 2013*; *Perea and Araque, 2010*; *Robel et al., 2011*). Further characterization of planarian glia, especially their developmental origin, will provide insight into the long-standing question of whether invertebrate and vertebrate glia share a common evolutionary origin (*Hartline, 2011*). Planarians are a tractable model organism that will be amenable to the study of glia in a highly regenerative member of the understudied Lophotrochozoan superphylum. We conclude that planarians possess glia and that the state of these cells localized within the CNS neuropil is regulated by midline Hh signaling.

## Materials and methods

### Planarian culture

Animals were maintained in 1x Montjuic planarian water at 20°C as previously described (*Sánchez Alvarado et al., 2002*). S2F1L3F2 sexual animals were used in dissection experiments and CIW4 asexual animals were used in all other experiments.

## Molecular biology

cDNA libraries of CIW4 planarian multi-stage total RNA were synthesized using SuperScript III (Thermo Fisher Scientific, Waltham MA). DNA fragments were amplified from cDNA with primers designed for Dresden Transcriptome Assembly sequences (*Liu et al., 2013*) and cloned into pGEM (Promega, Fitchburg, WI). For RNAi constructs, inserts were amplified from pGEM constructs and introduced using BP clonase (Thermo Fisher Scientific) into a Gateway vector containing flanking LacZ inducible promoters. Full-length sequences for *if-1* and *cali* were obtained with 5' and 3' RACE (Thermo Fisher Scientific).

## RNA interference

300ml of bacterial culture expressing dsRNA was pelleted and mixed with 1ml of 70% liver in planarian water as previously described (*Reddien et al., 2005a*). Asexual animals were fed 6 times at four-day intervals unless otherwise noted. Sexual animals were fed 12 times at four-day intervals. A gene not present in the planarian genome, *unc-22*, was used as a control in each RNAi experiment.

## Dissection

After four days of starvation, the animals were immersed in a 0.33N HCl solution for 30 s, washed once in PBS, washed once in PBS + 1% BSA, and immobilized dorsal-side up on a silicon elastomer pad with insect pins. One longitudinal incision and one lateral incision were made through the dorsal epidermis near the base of the pharynx. The epidermis was peeled away to expose the pharynx and a layer of gut tissue overlying the CNS. Collected tissue was placed immediately in Trizol Reagent (Thermo Fisher Scientific) and stored at −80C until all samples were processed.

## mRNA-seq analysis

cDNA libraries were generated with 1.0 ug total RNA from head fragments and 0.2 ug total RNA from dissected CNS samples using TruSeq RNA Library Preparation Kits v2 (Illumina, San Diego, CA). Libraries were prepared in duplicate with different index and sequenced Illumina HiSeq 2500. After read quality was assessed by FASTQC (RRID:SCR_005539), reads were mapped to the Dresden *S. mediterranea* Transcriptome Assembly (*Liu et al., 2013*) using Bowtie2 (RRID:SCR_005476) with the best single alignment reported and five bases trimmed from the 5' end to avoid misalignments due to index sequence contamination (*Langmead and Salzberg, 2012*). Read counts were determined from alignment data with Samtools (RRID:SCR_002105) (*Li et al., 2009*) and differential expression analysis was conducted with the DESeq2 R package (RRID:SCR_000154) (*Love et al., 2014*). Contigs with fewer than 100 reads per kilobase per million reads (RPKM) on average per condition were removed from further analysis to eliminate false positives, unless otherwise noted.

## Phylogenetic analysis

Gene families were predicted for each glial marker by BLASTX similarity with characterized human proteins and Interpro: Protein Sequence Analysis & Characterization (RRID:SCR_006695) (*Jones et al., 2014*). Amino acid sequences of family members from *Homo sapiens, Danio rerio, Drosophila melanogaster*, and *Caenorhabditis elegans* as well as hypothetical protein sequences from representative Deuterostome, Protostome, and Radiata species identified by BLASTX (RRID:SCR_001653) were aligned using MUSCLE (RRID:SCR_011812) with default parameters (*Kuo and Weisblat, 2011*). Poorly aligned segments were eliminated using GBlocks (*Castresana, 2000*). Phylogenetic trees were constructed using maximum likelihood analyses (PhyML) with the WAG amino acid substitution model and 1000 bootstrap replicates (*Guindon et al., 2010*). Resulting trees were visualized as cladograms using FigTree (RRID:SCR_008515).

## RNA in situ hybridization and immunofluorescence

RNA probes were synthesized with digoxigenin (Roche, Basel, Switzerland), fluorescein (Roche), or dinitrophenol nucleotides (PerkinElmer, Waltham, MA). For whole-mount in situ hybridization (WISH) and fluorescent in situ hybridization (FISH), animals were fixed in 4% formaldehyde according to published protocols (*Pearson et al., 2009*; *Scimone et al., 2016*). FISH protocols were followed as previously described using RNA probe dilutions at 1:1000, anti-digoxigenin peroxidase (Roche Cat#

11207733910, RRID:AB_514500) at 1:500, anti-fluorescein peroxidase (Roche Cat# 11426346910, RRID:AB_840257) at 1:300, and anti-dinitrophenol peroxidase at 1:100 (*Pearson et al., 2009*).

Rabbit polyclonal antibodies for IF-1 protein were raised against peptides with amino acid sequence 'TENNQIENSKEKTVC' (GenScript, Piscataway, NJ). For immunofluorescence, animals were fixed in Carnoy's fixative and stained as previously described (*Newmark and Sanchez Alvarado, 2000*; *Wenemoser and Reddien, 2010*). Anti-IF-1 antibody was used at 0.4 ug/ml, anti-Synapsin antibody (DSHB Cat# 3C11 (anti SYNORF1), RRID:AB_528479) at 1:1000, and anti-α-Tubulin (Lab Vision Cat# MS-581, RRID:AB_144075) at 1:1000, and were developed with tyramide signal amplification (Thermo Fisher Scientific).

To detect nuclei, animals were stained in DAPI overnight prior to mounting in VectaShield (Vector Labs, Burlingame, CA). Samples were imaged with an LSM 700 confocal microscope (Carl Zeiss AG, Oberkochen; Germany) and processed with Fiji/ImageJ (RRID:SCR_002285) (*Schindelin et al., 2012*). Cell counts for neuropil regions were normalized to cross-sectional area of the cephalic ganglia lobes. Cell counts for heads excluding the neuropil region were normalized to the cross-sectional area of the head.

## RNA probe specificity

RNA probe specificity for a target gene was determined by performing whole-mount in situ hybridization on animals following inhibition of the gene. Animals were fed one to four times with bacteria expressing dsRNA for a control gene or the target gene. After the last feeding, the animals were given five days to clear the intestine of lingering RNAi food prior to fixation.

## Expansion microscopy

Protein-retention expansion microscopy was performed according to published protocols (*Tillberg et al., 2016*) with minor adaptations for use with planarian tissue. Briefly, animals fixed with Carnoy's fixative and developed with tyramide signal amplification were treated overnight in 100 ug/ml acryloyl-X, SE (Thermo Fisher Scientific) in PBS at room temperature. Animals were subsequently embedded in polyelectrolyte gel and digested with 200 ug/ml Proteinase K (Thermo Fisher Scientific) overnight at room temperature. Following multiple washes in water, animals achieved >4-fold expansion along each axis. Samples were imaged by confocal microscopy (Leica SP8) with a 25x water immersion objective and processed with Imaris 8.3 (BitPlane, RRID:SCR_007370).

## Irradiation

Animals were exposed to 6000 rads of ionizing radiation (GammaCell) to ablate all dividing cells as previously described (*Wagner et al., 2011*). Treated animals were subsequently fed dsRNA-expressing bacteria three times at d0, d4, and d8. Animals were fixed immediately after onset of anterior regression at d11.

## Acknowledgements

We would like to thank E S Boyden for protein-retention expansion microscopy protocols, O Wurtzel, J van Wolfswinkel, and K Kravarik for support with RNA sequencing data analysis, M Srivastava for support with phylogenetic analysis, L E Cote for assistance with GEO, and members of the Reddien lab for comments and suggestions. We acknowledge NIH (R01GM080639) support. PWR is a Howard Hughes Medical Institute Investigator and an associate member of the Broad Institute of Harvard and MIT.

## Additional information

### Funding

| Funder | Grant reference number | Author |
| --- | --- | --- |
| Howard Hughes Medical Institute | | Peter W Reddien |
| National Institutes of Health | R01GM080639 | Peter W Reddien |

The funders had no role in study design, data collection and interpretation, or the decision to submit the work for publication.

## Author contributions

IEW, SWL, Conception and design, Acquisition of data, Analysis and interpretation of data, Drafting or revising the article; MLS, Acquisition of data, Analysis and interpretation of data, Drafting or revising the article; TRC, PWR, Conception and design, Analysis and interpretation of data, Drafting or revising the article

## Author ORCIDs

Thomas R Clandinin, http://orcid.org/0000-0001-6277-6849
Peter W Reddien, http://orcid.org/0000-0002-5569-333X

## Additional files

### Major datasets

The following dataset was generated:

| Author(s) | Year | Dataset title | Dataset URL | Database, license, and accessibility information |
|---|---|---|---|---|
| Wang IE, Lapan SW, Scimone ML, Reddien PW | 2016 | Gene expression profiling of planarian heads or cephalic ganglia after inhibition of Hedgehog signaling pathway genes by RNAi | www.ncbi.nlm.nih.gov/geo/query/acc.cgi?acc=GSE81059 | Publicly available at the NCBI Gene Expression Omnibus (accession no: GSE81059) |

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
