## [Decision Letter]

Thank you for submitting your article "Hedgehog signaling regulates gene expression in planarian glia-like cells" for consideration by *eLife*. Your article has been favorably evaluated by Fiona Watt (Senior Editor) and three reviewers, one of whom, Alejandro Sánchez Alvarado (Reviewer #1), is a member of our Board of Reviewing Editors..

The reviewers have discussed the reviews with one another and the Reviewing Editor has drafted this decision to help you prepare a revised submission.

Summary:

Here, Wang and colleagues report on the identification of 2 Hedgehog-pathway-modulated CNS genes (*if-1* and *cali*) which appear to be expressed in cells that exhibit glial-like morphology. These genes were identified via RNAseq and comparisons of the transcriptomes of isolated cephalic ganglia tissue and head fragments. The authors report that *if-1* and *cali* are expressed in an overlapping population of cells in the nervous system and that these cells: (I) do not express known neural markers, (II) they express a panel of neurotransmitter reuptake genes, and (III) using an *if-1* antibody they show these cells send out projections that appear to ensheath neurons. On the basis of these lines of evidence, the authors conclude that planarians possess glial cells and that these cells form part of this organism's CNS.

While the manuscript is of high quality, the following essential revisions should be addressed.

Essential revisions:

1) The data as presented appear insufficient to make glial cell-type distinctions. Two glial cell populations are postulated to exist: *if-1^+^/cali^+^* GLCs and *if-1^-^/cali^-^* GLCs. Given that the expression of *if-1* allows for the detection of the glial-like processes and that equivalent reagents do not exist for *if-1^-^/cali^-^* GLCs, it is still possible that these cells may not be glial cells proper but rather a precursor cell population. The authors must provide more evidence to support the initial claim for the existence of two glial cell types.

2) Similarly, the distinction between pGLCs and nGLCs seems unnecessary, and perhaps even misleading. How were the boundaries of the neuropil established for FISH experiments like the one in Figure 2? This was not explained. Figure 2 and Figure 5 suggest there are at least some *if-1/cali*-positive, Hh-responsive GLCs outside the neuropil, and some *if-1/cali*-negative GLCs inside the neuropil. Are these "real" results, or false-positives/negatives? In any case, the key distinction seems to be whether the cells are responding to Hh or not (regardless of location), and it is not yet clear whether this actually has any functional significance.

3) Given the author's claim that the *if-1/cali* positive cells may represent planarian glia, and based on the morphology reported, it would seem appropriate for the authors to look at the ultrastructure of these cells via electron microscopy (EM). Ideally, if the anti-*If-1* antibody were to work, perhaps immunoEM should be attempted. Alternatively, additional images such as those in Figure 4 would help support how representative the reported results may or may not be. Given the relative novelty of finding glial cells in planarians, EM efforts would help cement the notion of the existence of these cells in planarians.

4) The conclusion that "regionalized expression in the cephalic ganglia of several orthologs of Hh-responsive vertebrate neural tube transcription factors are unaffected by Hh signaling in planarians" (Discussion) may be somewhat overstated. This assumption is based solely on RNAi experiments in intact animals (understandable as the regenerating animals fail to form heads), leaving open the possibility that Hh signaling is required for the initial establishment of transcription factor expression domains during development and/or regeneration, but not for maintenance of these domains during cell turnover. It may be worth acknowledging this possibility in the Discussion section on the ancestral functions of Hh signaling in the CNS.

---

## [Author Response]

*Essential revisions:*

*1) The data as presented appear insufficient to make glial cell-type distinctions. Two glial cell populations are postulated to exist: if-1^+^/cali^+^ GLCs and if-1^-^/cali^-^ GLCs.*

We agree that glial cell-type distinctions cannot be made with available data, and that was not our intent. We are therefore happy for this opportunity to clarify wording about this topic in the paper. Our conclusion is that Hedgehog signaling induces expression of *if-1* and *cali*, rather than specifying one glial type among multiple. Whether expression of two genes and spatial restriction of a population is sufficient to define the population as a distinct cell type rather than a cell state is a discussion we have decided to avoid. We have added and modified text throughout the manuscript, such as in the Discussion section “Planarian glia gene expression is regulated by Hh signaling” to help clarify our main conclusion and to avoid wording that might imply a conclusion about multiple cell types. For example: “We currently do not have evidence supporting or rejecting the presence of multiple planarian glia cell types. It will be of interest to further investigate these possibilities by examining the function of planarian glia responsive to Hh signaling as well as planarian glia outside of the neuropil that do not express *if-1* or *cali* in *ptc(RNAi)* animals.”

*Given that the expression of if-1 allows for the detection of the glial-like processes and that equivalent reagents do not exist for if-1^-^/cali^-^ GLCs, it is still possible that these cells may not be glial cells proper but rather a precursor cell population. The authors must provide more evidence to support the initial claim for the existence of two glial cell types.*

We have also provided more evidence that the ectopic expression of *if-1* and *cali* in *ptc(RNAi)* animals is irradiation-insensitive, suggesting that the glia population outside the neuropil is unlikely to represent a precursor population. We show that *glut*_+_/*if-1*_-_/*cali*_-_ cells are unaffected by irradiation over a time scale that would be sufficient for precursor loss. Moreover, we show that induction or loss of *if-1* and *cali* by Hh signaling perturbation is quantitatively unaffected by irradiation. These data are now represented in Figure 6. We show that induction of *if-1* and *cali* in *ptc(RNAi)* animals occurred in existing (irradiation-insensitive) *glut*_+_ cells with branched morphology. We have substantially modified the section titled “Hh signaling promotes expression of *if-1* and *cali* in existing planarian glia” to state our conclusions more directly and incorporate new information.

*2) Similarly, the distinction between pGLCs and nGLCs seems unnecessary, and perhaps even misleading.*

We agree that naming planarian glia based on their localization is confusing and therefore we have abandoned this naming scheme in our manuscript. Planarian glia in these two locations are now referred to as “within the neuropil” and “outside the neuropil”.

*How were the boundaries of the neuropil established for FISH experiments like the one in Figure 2? This was not explained.*

To address this important concern, we have added a panel to Figure 2—figure supplement 1 that outlines the neuropil region in an animal labeled with *pc2* and DAPI. The legend includes a statement on how the boundaries of the neuropil are defined using DAPI staining and how cells are classified as either within the neuropil or outside the neuropil.

*Figure 2 and Figure 5 suggest there are at least some if-1/cali-positive, Hh-responsive GLCs outside the neuropil, and some if-1/cali-negative GLCs inside the neuropil. Are these "real" results, or false-positives/negatives? In any case, the key distinction seems to be whether the cells are responding to Hh or not (regardless of location), and it is not yet clear whether this actually has any functional significance.*

It is formally possible that *if-1*_-_/*cali*_-_ cells within the neuropil that express planarian glia markers are false negatives. However, we believe this possibility to be remote as these are strong probes and we generally observe strong labeling of positive cells within the neuropil. Moreover, there is a high degree of overlap between *if-1* and *cali* in double FISH (~95%), indicating a high efficiency of staining. We agree that the functional significance of *if-1/cali* expression in neuropil cells is unclear at present as we do not observe a phenotype by RNAi. We conclude that the planarian glia outside the neuropil that express *if-1* and *cali* are not false positives for two reasons: they strongly express both genes and they have elaborate morphologies similar to those cells found within the neuropil. We have modified Figure 2 to show cell morphologies by FISH for cells in the cephalic ganglia neuropil, the ventral nerve cord neuropil, and near the head rim. Corresponding changes to the text have been made in Results section “Expression and localization of *if-1* and *cali* is altered by Hh pathway perturbation”.

*3) Given the author's claim that the if-1/cali positive cells may represent planarian glia, and based on the morphology reported, it would seem appropriate for the authors to look at the ultrastructure of these cells via electron microscopy (EM). Ideally, if the anti-If-1 antibody were to work, perhaps immunoEM should be attempted. Alternatively, additional images such as those in Figure 4 would help support how representative the reported results may or may not be. Given the relative novelty of finding glial cells in planarians, EM efforts would help cement the notion of the existence of these cells in planarians.*

Unfortunately, our attempts at generating publication quality electron micrographs of the neuropil region that we are comfortable publishing have not been successful. We agree that immunoEM would be ideal for identifying the cells of interest in electron micrographs, but our anti-IF-1 antibody requires a dehydrating fixation condition that is generally thought to be incompatible with electron microscopy. We therefore took the reviewers’ suggestion to provide additional images like those in Figure 4 and to enhance our imaging analysis of these cells with development of a new method for planarian immunofluorescence. We have included more immunofluorescence images, providing more detail of IF-1_+_ processes throughout multiple regions of the planarian. First, we added Imaris 3D reconstructions of a synaptic glomerulus and an orthogonal branch from an animal labeled with IF-1 and Synapsin antibodies to Figure 4. Second, we added a new supplemental figure (Figure 4—figure supplement 2) that includes multiple planes of a confocal stack for six different regions of an animal labeled with IF-1 and Synapsin antibodies. Finally, we have adapted the recently published method for protein- retention expansion microscopy to planarians. This method expands fixed and immunolabeled tissue 64-fold in volume, allowing for both greater resolution with conventional confocal microscopy and increased clarity. Confocal stacks as well as 3D reconstructions for four different regions of an expanded planarian were added as Video 1–Video 8. With this higher resolution imaging, we were able to confirm the adjacency of IF-1_+_ processes to clusters of high synaptic density. A new Methods section was included to describe the protocol for expansion microscopy in planarians. We hope that these added figures help cement the elaborate morphology of these cells as well as their affiliation with synapses.

*4) The conclusion that "regionalized expression in the cephalic ganglia of several orthologs of Hh-responsive vertebrate neural tube transcription factors are unaffected by Hh signaling in planarians" (Discussion) may be somewhat overstated. This assumption is based solely on RNAi experiments in intact animals (understandable as the regenerating animals fail to form heads), leaving open the possibility that Hh signaling is required for the initial establishment of transcription factor expression domains during development and/or regeneration, but not for maintenance of these domains during cell turnover. It may be worth acknowledging this possibility in the Discussion section on the ancestral functions of Hh signaling in the CNS.*

We agree that our analysis of nervous system transcription factor expression patterns in *hh(RNAi)* and *ptc(RNAi)* animals was not conducted in regenerating animals and therefore, in the Discussion section “Ancestral roles of Hh signaling in CNS development”, we modified the text to more accurately state that regionalized expression did not change in uninjured planarians. We also comment on the possibility that this effect can only be observed in regenerating animals and propose future directions of study to indicate this is still an open question.